# The ALFA-tag is a highly versatile tool for nanobody-based bioscience applications

Hansjörg Götzke[1,10], Markus Kilisch[1,2,10], Markel Martínez-Carranza [3,10], Shama Sograte-Idrissi [4,5], Abirami Rajavel [1], Thomas Schlichthaerle[6,7], Niklas Engels [8], Ralf Jungmann [6,7], Pål Stenmark [3,9], Felipe Opazo [1,4,5] & Steffen Frey [1]

Specialized epitope tags are widely used for detecting, manipulating or purifying proteins, but often their versatility is limited. Here, we introduce the ALFA-tag, a rationally designed epitope tag that serves a remarkably broad spectrum of applications in life sciences while outperforming established tags like the HA-, FLAG®- or myc-tag. The ALFA-tag forms a small and stable α-helix that is functional irrespective of its position on the target protein in prokaryotic and eukaryotic hosts. We characterize a nanobody (NbALFA) binding ALFA-tagged proteins from native or fixed specimen with low picomolar affinity. It is ideally suited for super-resolution microscopy, immunoprecipitations and Western blotting, and also allows in vivo detection of proteins. We show the crystal structure of the complex that enabled us to design a nanobody mutant (NbALFA[PE]) that permits efficient one-step purifications of native ALFA-tagged proteins, complexes and even entire living cells using peptide elution under physiological conditions.

[1] NanoTag Biotechnologies GmbH, Rudolf-Wissell-Straße 28a, 37079 Göttingen, Germany. [2] Institute of Molecular Biology, University Medical Center Göttingen, Humboldtallee 23, 37073 Göttingen, Germany. [3] Department of Biochemistry and Biophysics, Stockholm University, S-106 91 Stockholm, Sweden. [4] Institute of Neuro- and Sensory Physiology, University Medical Center Göttingen, Humboldtallee 23, 37073 Göttingen, Germany. [5] Center for Biostructural Imaging of Neurodegeneration (BIN), University Medical Center Göttingen, Von-Siebold-Straße 3a, 37075 Göttingen, Germany. [6] Faculty of Physics and Center for Nanoscience, LMU Munich, Geschwister-Scholl-Platz 1, 80539 Munich, Germany. [7] Max Planck Institute of Biochemistry, Am Klopferspitz 18, 82152 Martinsried, Germany. [8] Institute of Cellular and Molecular Immunology, University Medical Center Göttingen, Humboldtallee 34, 37073 Göttingen, Germany. [9] Department of Experimental Medical Science, Lund University, Lund 221 00, Sweden. [10]These authors contributed equally: Hansjörg Götzke, Markus Kilisch, Markel Martínez-Carranza. Correspondence and requests for materials should be addressed to F.O. (email: fopazo@gwdg.de) or to S.F. (email: steffen.frey@nano-tag.com)

Epitope tags are employed in virtually any aspect of life sciences[1,2]. They are used in biotechnology to facilitate the expression and purification of recombinant proteins[1] or in cell biology to monitor the biogenesis or spatial organization of a given protein of interest (POI)[3,4]. Other uses include, e.g., the immunoprecipitation of protein complexes studied by mass spectrometry[5,6], or protein manipulations using tag-binding reagents in living cells[7,8].

While a given tag might be ideal for a specific application, it may completely fail in others. This is a result of how the tags have been generated – typically as byproducts while developing antibodies against specific POIs (for example the myc-tag[9], the HA-tag[10] or the Spot-tag®[11,12]). Other tags, like the His-tag[13], have been rationally designed for a specific application. None of the available tags covers the full range of current biological applications (see Supplementary Table 1 for details). For example, the His-tag provides poor results in immunostaining and imaging applications, albeit it is excellent for protein purification. The FLAG®-, myc- and HA-tags have often been used for immunostainings, but due to the large size of the antibodies used as binders, they are suboptimal for super-resolution microscopy and exceedingly difficult to express within cells. These tags therefore cannot be used to reveal or manipulate POIs in living cells. Similar considerations apply for the Twin-Strep-tag® (TST)[14], which is in addition fixation-sensitive, and therefore not suitable for immunostainings.

More recently the EPEA-tag[15] (also known as C-Tag) and the Spot-tag®[11,12] have been identified as tags recognized by camelid single-domain antibodies (sdAbs, also known as nanobodies[16]). In contrast to conventional antibodies, sdAbs are monomeric, small, show an enhanced performance in super-resolution microscopy[17,18] and can, in principle, even be used for intracellular applications[7,8]. Unfortunately, both of these systems have several problems that override the potential advantages given by their nanobody binders. For instance, both nanobodies detect endogenous proteins (α-synuclein and β-catenin, respectively) and display comparably poor affinities when used as monovalent binders (Supplementary Table 1) implying that they are suboptimal for advanced microscopy applications or pull-downs of low abundant proteins. Additionally, both tags have so far not been reported to work in living cells, and the EPEA-tag can only be used at the C-terminus of target proteins.

To overcome the limitations of the available epitope tags, a first step is to define a set of desired features. A truly versatile tag should not affect the structure, topology, localization, solubility, oligomerization status or polar interactions of the tagged protein[19,20]. It should therefore be small, monomeric, highly soluble[21] and electroneutral. For highest versatility, it should in addition be resistant to chemical fixation. To avoid specific background signals, the optimal tag should be unique in eukaryotic and prokaryotic hosts, while being well expressed and protease resistant. Similarly, the molecule binding such tag should fulfill certain characteristics. It should not only be small for an optimal access to crowded regions and have a minimal linkage error in super-resolution microscopy[22,23], but also specifically bind the tag with high affinity. For in vivo imaging and manipulations, the binder should be genetically accessible and fold properly within various host organisms. For biochemical applications, the preferred binder should allow for both, stringent washing and native elution of immunoprecipitated proteins or complexes. Strikingly, currently existing epitope tag systems fail to fulfill the complete set of mentioned criteria (Supplementary Table 1). To manufacture an epitope tag system with ultimate versatility, the only way is to design it de novo.

We now introduce the ALFA-tag, which addresses these clear objectives. Its 15 amino acid sequence is hydrophilic, uncharged at

physiological pH and devoid of residues targeted by amine-reactive fixatives or cross-linkers. It has a high propensity to form a stable α-helix that spontaneously refolds even after exposure to harsh chemical treatment. Due to its compact structure, the tag is physically smaller than most linear epitope tags. The ALFA-tag is compatible with protein function and can be placed at the N- or C-terminus of a POI, or even in between two separately folded domains.

As a counterpart binding the ALFA-tag, we additionally introduce a high-affinity nanobody (NbALFA), which proves to be suitable for super-resolution imaging and intracellular detection of ALFA-tagged proteins and allows very efficient and clean immunoprecipitations and Western blots. However, it is virtually impossible to separate NbALFA from the ALFA-tag under native conditions, hindering its application for the purifications of native protein complexes, organelles or cells. Based on the crystal structure of the NbALFA-ALFA peptide complex we engineered a nanobody variant (NbALFA^PE; for Peptide-Elution) suitable for highly specific purification of ALFA-tagged proteins, protein complexes and even entire living cells under physiological conditions. The rationally designed ALFA system presented here serves a remarkably broad range of applications from biotechnology to cell biology. A single tag can therefore replace a great variety of traditional epitope tags.

## Results

**The ALFA system.** The sequence of the minimal ALFA-tag (SRLEEELRRRLTE; Fig. 1a) is inspired by an artificial peptide (SRLEEELRRRL) reported to form a stable α-helix in solution[24]. It was selected based on the following criteria: (i) It features a high alpha-helical content, (ii) The sequence is absent in common eukaryotic model systems, (iii) It is hydrophilic and neutral at physiological pH while retaining moderate hydrophobic surfaces and (iv) It does not contain any primary amines that are modified by aldehyde-containing fixatives. The additional Thr-Glu (TE) dipeptide was added to the original peptide to neutralize its positive net charge and thus fully comply with the selection criteria defined above. To minimize any potential influence of neighboring secondary structures, the minimal ALFA-tag sequence was in addition framed by prolines (Fig. 1a).

In order to develop nanobodies binding the ALFA-tag, we immunized alpacas and selected nanobodies specifically targeting ALFA-tagged proteins by a nanobody selection method (see Online Methods) developed in-house. The best binder (NbALFA; Fig. 1b) was genetically modified with ectopic cysteines allowing for a site-specific immobilization or fluorophore attachment[25].

**Detection of ALFA-tagged proteins by immunofluorescence.** In immunofluorescence (IF) applications on PFA-fixed mammalian cells, fluorescently labeled NbALFA specifically recognized target proteins harboring ALFA-tags at different locations within the proteins (Supplementary Fig. 1). Importantly, all tested proteins showed their characteristic localization (Tom70-EGFP-ALFA: mitochondrial outer membrane; ALFA-vimentin: filamentous structures; EGFP-ALFA-TM: plasma membrane), the ALFA-tag was therefore compatible with the folding and proper localization of the tagged proteins. In a quantitative assay, the nucleocytoplasmic distribution of EGFP carrying an ALFA-tag at either terminus was indistinguishable from non-tagged EGFP (Supplementary Fig. 2). An atypical interaction to cellular membranes or organelles was not observed. We therefore concluded that the ALFA-tag seems not to impair the physiological behavior of the tagged POIs.

**Resistance to amine-reactive fixatives.** In order to facilitate optimization of fixation conditions, it is advantageous if a given

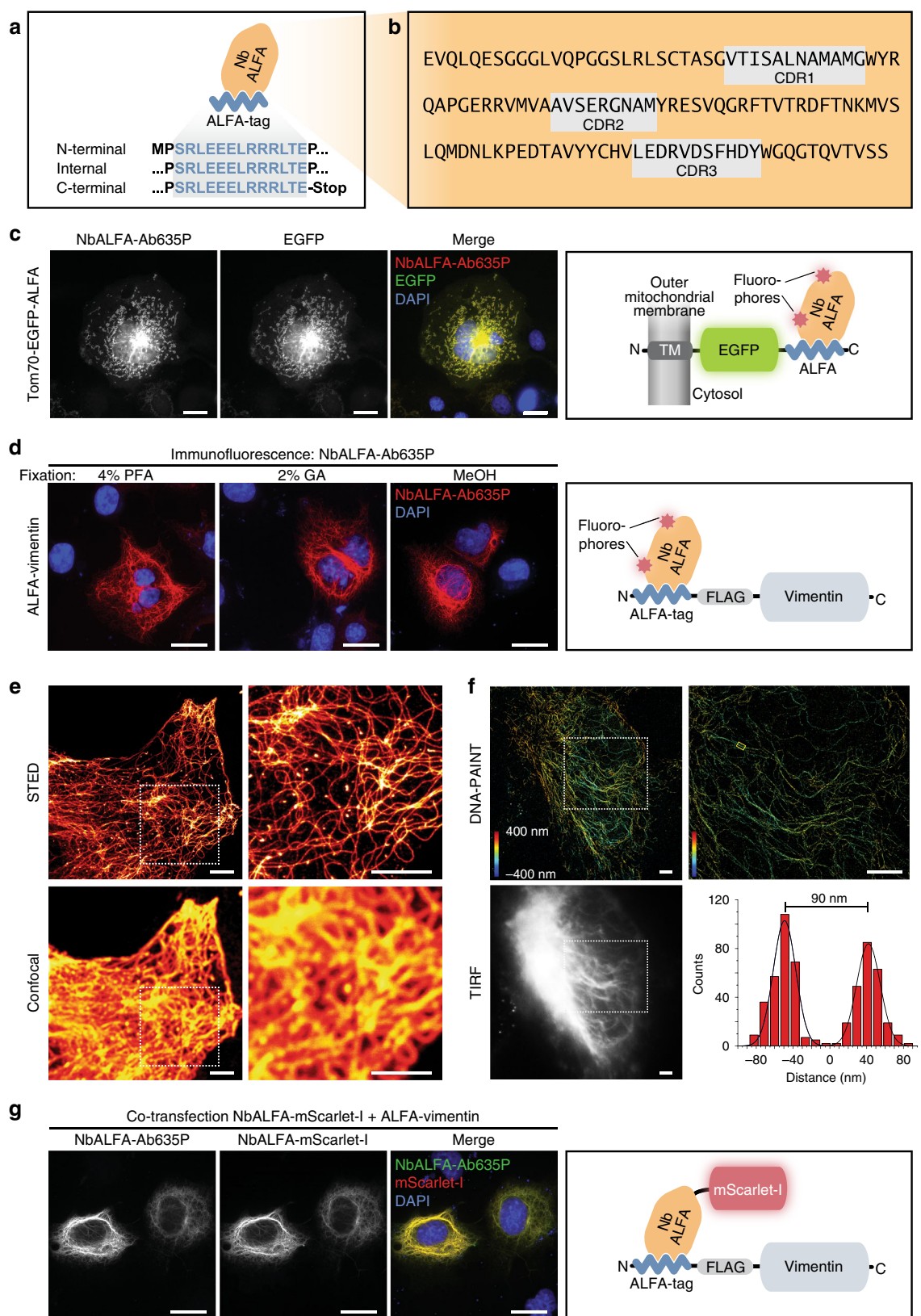

**a**

ALFA-tag

| N-terminal | **MP**SRLEEELRRRLTE**P**... |
| Internal | ...**P**SRLEEELRRRLTE**P**... |
| C-terminal | ...**P**SRLEEELRRRLTE-**Stop** |

**b**

EVQLQESGGGLVQPGGSLRLSCTASG VTISALNAMAMG WYR
CDR1

QAPGERRVMVA AVSERGNAM YRESVQGRFTVTRDFTNKMVS
CDR2

LQMDNLKPEDTAVYYCHV LEDRVDSFHDY WGQGTQVTVSS
CDR3

**c** Tom70-EGFP-ALFA

NbALFA-Ab635P    EGFP    Merge

NbALFA-Ab635P
EGFP
DAPI

Outer mitochondrial membrane — Fluoro-phores — NbALFA
N — TM — EGFP — ALFA — C
Cytosol

**d** ALFA-vimentin

Immunofluorescence: NbALFA-Ab635P

Fixation:    4% PFA    2% GA    MeOH

NbALFA-Ab635P
DAPI

Fluoro-phores — NbALFA
N — ALFA-tag — FLAG — Vimentin — C

**e**

STED

Confocal

**f**

DNA-PAINT

400 nm
−400 nm

TIRF

90 nm

Counts
120
80
40
0
−80  −40   0   40  80
Distance (nm)

**g** Co-transfection NbALFA-mScarlet-I + ALFA-vimentin

NbALFA-Ab635P    NbALFA-mScarlet-I    Merge

NbALFA-Ab635P
mScarlet-I
DAPI

NbALFA — mScarlet-I
N — ALFA-tag — FLAG — Vimentin — C

epitope tag is compatible with various fixation procedures. In contrast to many established epitope tags (Supplementary Table 1), the ALFA-tag does not contain lysines by design. Consequently, it could be detected after standard fixation with paraformaldehyde or methanol, and was even resistant to 2% glutaraldehyde (Fig. 1d). The ALFA-tag is thus compatible with

standard fixation methods and has the potential to be employed in electron microscopy, where glutaraldehyde is preferred due to its superior preservation of cellular structures.

**Super-resolution microscopy.** Due to the small size of both the ALFA-tag and NbALFA, the ALFA system results in a minimal

**Fig. 1** Nanobody-based detection of ALFA-tagged proteins in immunofluorescence applications. **a** Sketch of NbALFA bound to the ALFA-tag. Given are ALFA sequences for tagging at various positions. **b** Sequence of NbALFA. Gray boxes indicate CDRs 1–3 (complementarity determining regions 1–3; AbM definition). **c** COS-7 cells transfected with Tom70-EGFP-ALFA were fixed with paraformaldehyde (PFA) and stained with NbALFA coupled to AbberiorStar635P (NbALFA-Ab635P). Left to right: NbALFA-Ab635P; intrinsic EGFP signal; overlay incl. DAPI stain; sketch illustrating the detection of Tom70-EGFP-ALFA. Scale bars: 20 μm. **d** ALFA-vimentin was detected with NbALFA-Ab635P after fixation with 4% PFA, 2% glutaraldehyde (GA), or 100% Methanol (MeOH). Scale bars: 20 μm. **e** STED and confocal images of COS-7 cell transiently transfected with ALFA-vimentin and stained with NbALFA-Ab635P. Color scheme: Red Hot (ImageJ). Scale bars: 2.5 μm. **f** HeLa cells transfected with ALFA-vimentin were stained with NbALFA bearing a 10-nucleotide single stranded DNA before imaging by 3D DNA-PAINT. Scale bars: 2.5 μm. The histogram refers to a region (small yellow rectangle) where 2 vimentin filaments are resolved although being only ~90 nm apart. The localization precision was 5.2 nm. **g** COS-7 cells were co-transfected with an NbALFA-mScarlet-I fusion and ALFA-vimentin. NbALFA-mScarlet-I co-localizes with ALFA-vimentin detected by immunofluorescence using NbALFA-Ab635P. This shows that NbALFA expressed in the cytoplasm of mammalian cells can be used for targeting ALFA-tagged proteins in living cells. Scale bars: 20 μm. N: N-terminus; C: C-terminus, TM: transmembrane domain. Colors scheme used for sketches: NbALFA (orange), ALFA-tag (blue), GFP (green), mScarlet-I (red), fluorophore (red star), FLAG®-tag (gray), vimentin (light blue)

linkage error and is thus perfectly suited for super-resolution microscopy. As examples, we imaged cells transfected with ALFA-vimentin by either STED microscopy[17] or 3D DNA-PAINT[26] using NbALFA directly coupled to a STEDable fluorophore or a short single-stranded oligonucleotide, respectively (Fig. 1e, f). Our results show that the ALFA system is compatible with the demanding conditions of these fluorescent super-resolution microscopy techniques.

**Detecting and manipulating ALFA-tagged proteins in vivo.** Some nanobodies are functional within mammalian cells and can thus be used to detect or manipulate target proteins in living cells[8,27]. Indeed, when co-expressing ALFA-tagged target proteins and NbALFA fused to mScarlet-I[28] in mammalian cells, NbALFA-mScarlet-I robustly co-localized with ALFA-tagged target proteins with minimal off-target signals, resulting in crisp detection of ALFA-tagged structures (Fig. 1g, Supplementary Fig. 3). This demonstrates the ability of NbALFA to bind ALFA-tagged proteins in living cells. This feature is very attractive to be used in combination with genome editing tools like CRISPR-Cas[29] and allows manipulation of ALFA-tagged proteins in vivo at their endogenous levels.

**Western blot.** We next tested the ALFA system in Western blot applications and could specifically detect ALFA-tagged vimentin in lysates from transfected cells using NbALFA labeled with IRDye800CW (Fig. 2a, Supplementary Fig. 4a). To compare the performance of the ALFA system with common epitope tags, we employed a fusion protein harboring HA-, myc-, FLAG®- and ALFA-tags (Fig. 2b). At identical concentrations of primary antibody or nanobody, the obtained ALFA-tag signal was 3–10-fold stronger than the signal obtained for all other epitope tags (Fig. 2c, Supplementary Fig. 4b). This is striking, since the ALFA-tag signal exclusively relied on the NbALFA fluorophores, while the signal of all other epitope tags was amplified using polyclonal secondary antibodies. Without further optimization, NbALFA yielded a remarkably linear signal over at least three orders of magnitude (Fig. 2d) and was able to detect as little as 100 pg of target protein. The detection limit was thus ~10-times better compared to all other epitope tags.

**Structure of the NbALFA bound to the ALFA peptide.** To better understand the interaction between NbALFA and the ALFA-tag, we solved the crystal structure of NbALFA in complex with the ALFA peptide (Fig. 3, Supplementary Fig. 5, Table 1). NbALFA adopts a canonical IgG fold[30] comprising two β-sheets connected by a disulfide bond. The paratope accepting the ALFA peptide extends from the nanobody's N-terminal cap to the side formed by the five-stranded β-sheet. It is mainly built from complementarity determining regions (CDRs[16]) 2 and 3 and,

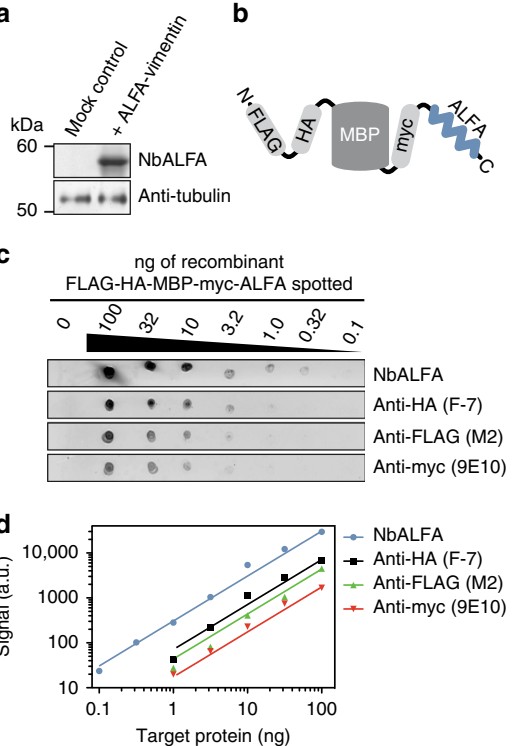

**Fig. 2** Detection of ALFA-tagged target proteins by fluorescent Western blot. **a** Lysates from COS-7 cells transfected with ALFA-vimentin or a mock control plasmid were analyzed by SDS–PAGE and Western blot. The blot was developed with NbALFA directly coupled to IRDye800CW or a mouse anti-tubulin primary antibody followed by FluoTag-X2 anti-Mouse IRDye680RD. Complete lanes including molecular weight markers are shown in Supplementary Fig. 4a. **b** Sketch of the *E. coli* maltose-binding protein (MBP) simultaneously fused to FLAG®-tag (FLAG), HA-tag (HA), myc-tag (myc) and ALFA-tag (ALFA). The multi-tag fusion protein was used for the experiments shown in **c**, **d**. ALFA-tag is shown in blue. **c** Dilution series of the protein sketched in **b** were spotted onto nitrocellulose membranes. Established monoclonal antibodies (M2, 9E10 and F-7) were used together with a goat anti-mouse secondary antibody coupled IRDye800CW to detect the FLAG®-, myc- and HA-tags, respectively. The ALFA-tag was detected using NbALFA coupled to IRDye800CW. The complete experiment with controls is shown in Supplementary Fig. 4b. **d** Double-logarithmic plot showing quantification of signals obtained in **c** in arbitrary units (a.u.) versus the amount of spotted target protein. Lines represent linear fits to the obtained values. Even without signal amplification by a secondary antibody, signals obtained using NbALFA were 3- to >10-times stronger than by established reagents recognizing the other epitope tags. At the same time, detection with NbALFA was 10-fold more sensitive and showed an excellent linearity over approximately three orders of magnitude

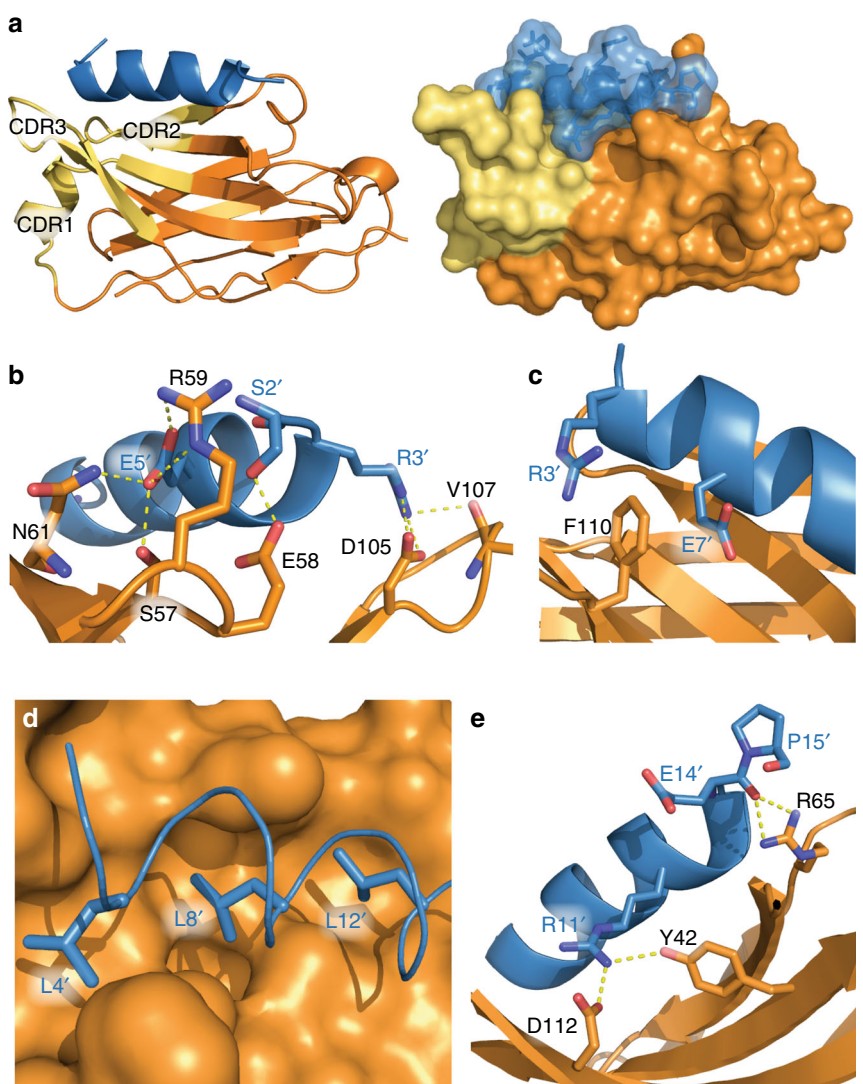

**Fig. 3** Structure of the NbALFA-ALFA peptide complex. **a** View on the NbALFA-ALFA peptide complex illustrated as cartoon (left) or surface representation (right). NbALFA: orange with CDRs 1–3 colored in yellow. ALFA peptide in blue. For both molecules, the N-terminal is oriented left, the C-terminal right. **b** Polar interactions within the N-terminal region of the ALFA peptide. ALFA peptide residues are denoted by an apostrophe. S2' and E5' form hydrogen bonds with CDR2 (S57, E58, R59 and N61). R3' reaches out to CDR3 and interacts with the backbone of V107 and the side chain of D105. **c** R3' and E7' sandwich F110 on the nanobody, forming a cation-Pi interaction (reviewed in ref. [52]). **d** Illustration of a hydrophobic cluster (L4', L8' and L12') facing the nanobody's hydrophobic cavity. **e** Polar interaction near the C-terminal of the ALFA peptide. The backbone of E14' forms hydrogen bonds with R65 while the side chain of R11' interacts with D112 and Y42 on the five-stranded β-sheet. Interestingly, Y42 has been described as a conserved residue in nanobodies of this particular architecture[8]

interestingly, also involves residues within the five-stranded β-sheet forming a hydrophobic cavity. As a result, the ALFA peptide is oriented parallel to the central axis of NbALFA. The ALFA peptide forms an α-helical cylinder with a length of ~2 nm and a diameter of ~1.3 nm, that is stabilized by a complex network of intramolecular interactions (Supplementary Table 2). In addition, the peptide forms multiple polar and hydrophobic contacts with NbALFA (Fig. 3b–e, Supplementary Table 3).

**Capture of ALFA-tagged proteins using ALFA Selector resins**. For biochemical purifications, we site-specifically immobilized NbALFA on an agarose-based resin (Fig. 4a). The resulting resin efficiently and strongly bound an ALFA-tagged GFP variant (shGFP2[31]): Even after 1 h competition with an excess of free ALFA peptide at 25 °C, most of the target protein remained on the resin (Fig. 4b, red solid line). In line with these observations,

SPR assays indicated that the affinity of NbALFA to an ALFA-tagged target was ~26 pM (Supplementary Fig. 6). We therefore called the NbALFA-charged resin ALFA Selector^ST (for Super-Tight). To allow for an efficient competitive elution of ALFA-tagged target proteins, we intended to weaken NbALFA. For that, we followed a rational approach based on the NbALFA-ALFA peptide structure: NbALFA variants featuring single or combined amino acid exchanges in spatial proximity to the ALFA peptide were individually tested for their binding and elution properties. The successfully weakened binder NbALFA^PE (for Peptide Elution) carries several mutations with respect to NbALFA, thereby removing specific interactions of NbALFA with the ALFA peptide. As a consequence, the affinity of NbALFA^PE to a fusion protein harboring a C-terminal ALFA-tag was reduced to ~11 nM (Supplementary Fig. 6). As intended, a NbALFA^PE-charged resin (ALFA Selector^PE) efficiently and stably captured shGFP2-ALFA while allowing an efficient release within ~15–20 min by

**Table 1 Data collection and refinement statistics**

| | NbALFA/ALFA peptide complex |
|---|---|
| Data collection | |
| Space group | C 1 2 1 |
| Cell dimensions | |
| $a$, $b$, $c$ (Å) | 101.24, 32.38, 64.73 |
| $\alpha$, $\beta$, $\gamma$ (°) | 90.00, 147.22, 90.00 |
| Resolution (Å) | 1.5 |
| $R_{sym}$ or $R_{merge}$ | 0.063 (0.225) |
| $I/\sigma I$ | 12.70 (4.26) |
| Completeness (%) | 95.2 (94.0) |
| Redundancy | 5.6 (5.4) |
| Refinement | |
| Resolution (Å) | 50.33–1.5 |
| No. reflections | 17569/865 |
| $R_{work}/R_{free}$ | 0.165/0.200 |
| No. atoms | |
| Protein | 1067 |
| Ligand/ion | 28 |
| Water | 118 |
| B-factors | |
| Protein | 11.31 |
| Ligand/ion | 25.68 |
| Water | 18.56 |
| R.m.s. deviations | |
| Bond lengths (Å) | 0.0129 |
| Bond angles (°) | 1.857 |

competition with free ALFA peptide (Fig. 4b, c). Similar elution kinetics were found when the ALFA-tag was placed between two folded domains, while elution of ALFA-shGFP2 from ALFA Selector[PE] was slightly quicker (Supplementary Fig. 7). Remarkably, in the absence of competing peptide, spontaneous elution of all target proteins from both ALFA Selector variants was insignificant (Fig. 4b, c and Supplementary Fig. 7).

**Stringent washing and pH resistance**. We next performed stringent washing steps on both ALFA Selector variants bound to either ALFA-shGFP2 or shGFP2-ALFA (Fig. 4d). At 25 °C, all substrate–resin interactions were completely resistant to significant concentration of salt, Guanidinium-HCl, non-denaturing detergents or reducing agents. A small fraction of substrate was released from ALFA Selector[PE], but not from ALFA Selector[ST], upon treatment with up to 6 M urea.

In a similar assay, the loaded ALFA Selector resins were exposed to buffers adjusted to various pH (Fig. 4e). The interaction was stable at pH7.5 or 9.5, and only slightly affected at pH4.5. However, even after neutralization, both ALFA Selector resins remained completely non-fluorescent when washed with 100 mM Glycin at pH2.2. The eluted material, in contrast, successfully recovered its fluorescence at neutral pH, indicating that acidic elution with Glycin at pH2.2 is possible even from ALFA Selector[ST].

**Pull-down of ALFA-tagged proteins from complex lysates**. To address the specificity of the ALFA Selector resins in native pull-downs, we spiked E. coli or HeLa lysates prepared in PBS with recombinant ALFA-shGFP2 (Fig. 5a, left lane). The target protein specifically bound to both ALFA Selector variants but not to a control resin without coupled nanobody. As expected, ALFA-shGFP2 efficiently eluted from ALFA Selector[PE] using 200 μM of ALFA peptide in PBS. In contrast, significant elution from ALFA Selector[ST] was only observed upon treatment with SDS sample buffer. Importantly, pull-downs from both, E. coli (Fig. 5a) and

HeLa lysates (Fig. 5b), were highly specific: After peptide elution from ALFA Selector[PE] essentially all visible bands could be attributed to the input protein, and even in the SDS eluate, the number and strength of detectable impurities originating from lysate proteins were very low.

In a more delicate co-immunoprecipitation experiment, we tried to pull-down the E. coli YfgM-PpiD inner membrane protein complex[32] under native conditions (Fig. 5c). For this, either wild-type YfgM or YfgM–ALFA was expressed in a ΔyfgM strain under the control of its endogenous promoter[32]. ALFA Selector[PE] was able to specifically pull-down the YfgM–PpiD complex in a detergent-resistant manner from the lysate containing YfgM–ALFA. The native membrane protein complex could be recovered from ALFA Selector[PE] by peptide elution under physiological conditions showing that the ALFA-tag together with ALFA Selector[PE] resin is perfectly suited for native pull-downs of challenging (membrane) protein complexes.

**Isolation of live lymphocytes**. An envisioned application for the ALFA Selector[PE] is the specific enrichment of cells under physiological conditions. This may be particularly interesting e.g., for the generation of chimeric antigen receptor-modified T (CAR-T) cells, the precursors of which are usually obtained from blood[33]. To investigate if the ALFA system can be applied to enrich live blood cells, human peripheral blood mononuclear cells (PBMCs) were passed through an ALFA Selector[PE] column pre-charged with an ALFA-tagged nanobody recognizing CD62L, a surface marker typically present on naive T cells[34] (Fig. 6a). After washing, bound cells were eluted using ALFA peptide, stained with antibodies recognizing CD62L, the pan T cell marker CD3 and the pan B cell marker CD19, and analyzed by FACS (Fig. 6). Total PBMCs served as a control. Using this strategy, CD62L+ lymphocytes were enriched from 71.8 to 97.7% (Fig. 6b). In addition, we confirmed that the vast majority of ALFA peptide-eluted cells were CD3-positive T cells, while B cells represented a minor population of the isolated cells (Fig. 6c).

## Discussion

In the current study, we report the characterization of the ALFA system comprising the ALFA-tag, a highly versatile epitope tag, and two dedicated nanobodies. The primarily rational approach allowed us to equip the ALFA system with favorable features for a broad spectrum of applications. When selecting the ALFA-tag sequence, we not only made sure that the tag is small, monovalent, devoid of lysines, hydrophilic without carrying any net charge and absent within the proteome of relevant model organisms, but also that it would adopt a stable α-helical structure in solution. As a result, the ALFA-tag is by design highly specific (Fig. 5), insensitive to amine-reactive fixatives (Fig. 1c) and well tolerated by the tagged target proteins (Fig. 1 and Supplementary Figs. 1–3). The ALFA-tag could thus even be used for purification of a labile membrane protein complex (Fig. 5c). However, as for any other tag, specific effects on given target proteins have to be analyzed on a protein-to-protein basis.

The ALFA-tag with its comprehensive feature set contrasts to existing tags, which mostly fail to fulfill one or more parameters (Supplementary Table 1). For instance, large structured tags like GFP can affect the tagged protein localization and function[4,19]. On the other hand, intrinsically disordered small tags can adopt multiple conformations with unpredictable effects. The HA-tag, e.g., can be cleaved by caspases in mammalian cells[35]. In the worst case, tags may even target the POI for protein degradation[36]. Aldehyde-containing fixatives used for light and electron microscopy modify lysine-containing epitope tags like the myc-tag, which may impair antibody-based detection. Other tags (e.g.,

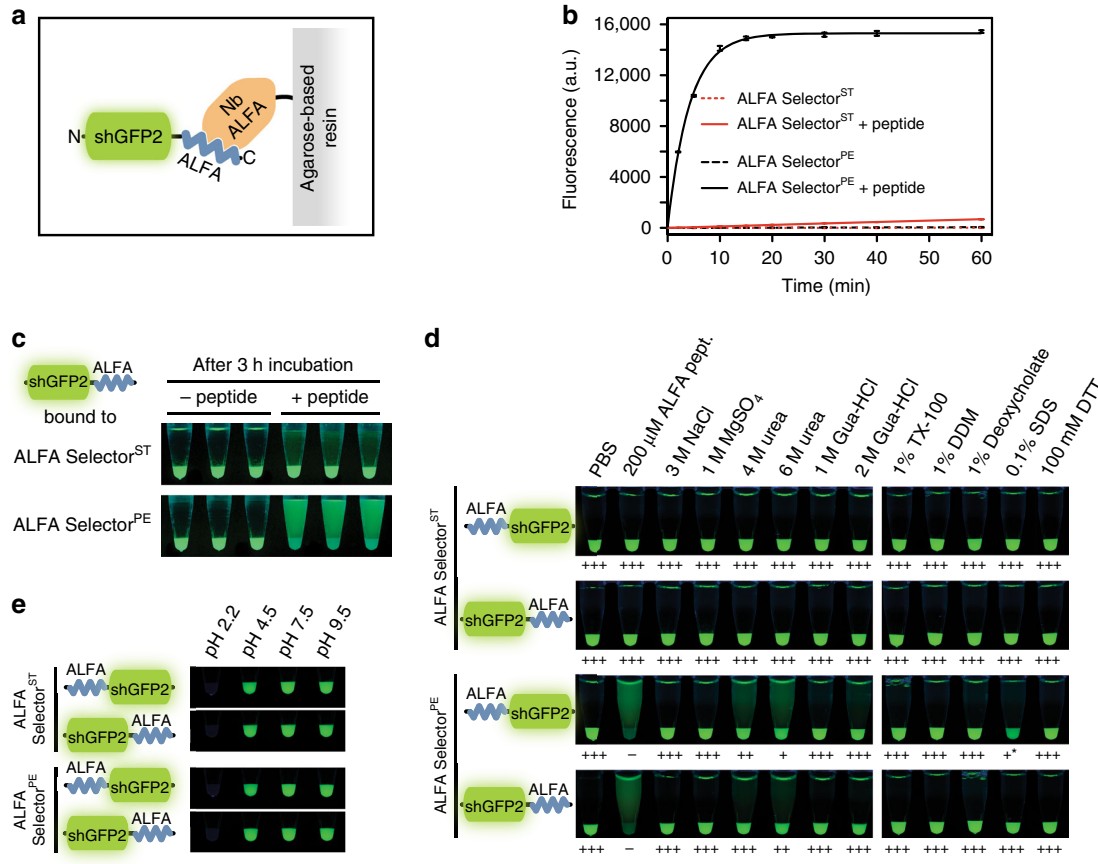

**Fig. 4** Immunoprecipitation of ALFA-tagged proteins using ALFA Selector resins. **a** Sketch of shGFP2-ALFA bound to a resin coupled to NbALFA (orange). **b**, **c** ALFA Selector^ST or ALFA Selector^PE were charged with shGFP2 harboring a C-terminal ALFA-tag. To estimate off-rates, the resins were incubated with an excess of free ALFA peptide at 25 °C. Control reactions were carried out without peptide. shGFP2 released from the resin was quantified using its fluorescence. The graph **b** shows mean fluorescence readings, as well as standard deviations (error bars; $n = 3$). Lines represent fits to a single exponential. A photo was taken upon UV illumination after 3 h of elution (**c**). **d** Resistance towards stringent washing steps. Both ALFA Selector variants were charged with either ALFA-shGFP2 or shGFP2-ALFA and incubated with a 10-fold volume of the indicated substances for 1 h at 25 °C with shaking. Without further washing steps, photos were taken upon UV illumination after sedimentation of the beads. A semi-quantitative evaluation of binding strengths is given below the images: Triple plus – very strong, double plus – strong, single plus – weak, minus – not detectable, asterisk – GFP fluorescence partially impaired by SDS. **e** Resistance towards various pH: Similar to **d**, here, however, the resins were washed to remove non-bound material after incubating at indicated pH for 30 min. Photos were taken after re-equilibration in PBS to allow for recovery of the GFP fluorescence. Color code used in sketches: NbALFA (orange), shGFP2 (green), ALFA-tag (blue)

the FLAG®-tag) carry significant net charges and may thus affect the POI's physiological electrostatic interactions. Tags like the myc-tag[9], the HA-tag[10], the Spot-tag®[11,12], the C-tag[15] and the Inntags[37] are derived from proteins present in model organisms. Therefore, the respective binders can also recognize the endogenous host proteins by default. In contrast to the C-tag[15] that only works at a POI's C-terminal, the ALFA-tag works at all accessible locations within the target protein without interfering with the protein's localization or topology (Supplementary Figs. 1–3).

For binding the ALFA-tag we developed nanobodies[16], because they are small, monovalent and robust, can easily be modified by genetic means and recombinantly produced in various organisms. They can therefore be site-specifically immobilized or quantitatively modified with fluorophores[25] or oligonucleotides. In comparison to conventional antibodies, nanobodies are thus superior for conventional and advanced microscopy[11,18] (Fig. 1e, f) since they can find more target epitopes in crowded areas[23], localize the fluorophore closer to the target protein and avoid artificial clustering of the POI[17,22]. Interestingly, NbALFA folds even within eukaryotic cells (Fig. 1g) and can thus be used as an intrabody[7] for detecting or manipulating ALFA-tagged proteins in vivo[7,8,27].

Nanobodies often fail to detect proteins in Western blot applications as they typically recognize three-dimensional epitopes. NbALFA, however, enables highly sensitive Western blot detection, suggesting that the ALFA-tag's α-helical structure refolds efficiently after SDS removal. Despite its monovalent binding mode and without further signal amplification, NbALFA significantly outperformed established anti-epitope tag tools regarding absolute signal intensity and detection limit (Fig. 2). We expect a better performance also in other high sensitivity applications like ELISA or microarray assays. Due to its resistance to amine-reactive fixatives (Fig. 1d), we even envision applying the ALFA system for immuno-electron microscopy and nanoSIMS[38].

The high-affinity binding of NbALFA to the ALFA-tag could be explained after solving the crystal structure of the NbALFA-ALFA peptide complex (Fig. 3). First, the peptide binds to NbALFA in an α-helical conformation and has an unusually high propensity to form a stable α-helix also in solution[24]. This may be explained by multiple intramolecular side-chain interactions within the peptide (Supplementary Table 2). As a result, no energy needs to be spent in forming the required structure during binding, which would otherwise disfavor complex

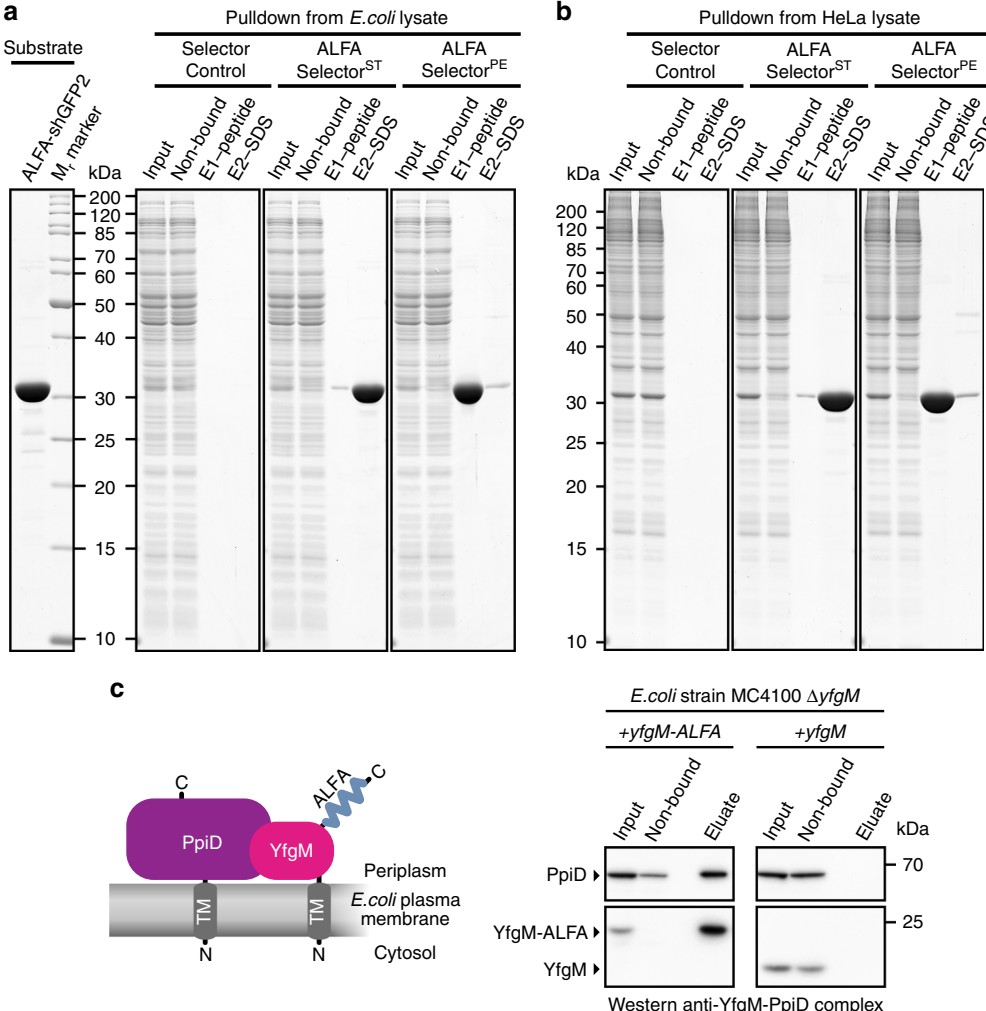

**Fig. 5** One-step affinity purifications using ALFA Selector resins. **a, b** E. coli (**a**) or HeLa (**b**) lysates blended with purified ALFA-tagged shGFP2 (**a**, left lane) were incubated with ALFA Selector[ST], ALFA Selector[PE] or an analogous resin without immobilized sdAb (Selector Control). After washing with PBS, the resins were incubated with 200 μM ALFA peptide for 20 min (E1–peptide) before eluting remaining proteins with SDS sample buffer (E2–SDS). Indicated fractions were analyzed by SDS–PAGE and Coomassie staining. Eluate fractions correspond to the material eluted from 1 μL of resin. **c** Native pull-down of an E. coli inner membrane protein complex. Left: Sketch of the target protein complex. Right: Detergent-treated lysates from a ΔyfgM strain complemented with either C-terminally ALFA-tagged (left panel) or untagged YfgM (right) were incubated with ALFA Selector[PE]. After washing with PBS, bound proteins were eluted using 200 μM ALFA peptide. Samples corresponding to 1/800 of the input and non-bound material or 1/80 of eluate fractions were resolved by SDS page and analyzed by Western blot. ALFA Selector[PE] specifically immunoprecipitated the native protein complex comprising ALFA-tagged YfgM and its interaction partner PpiD. In the control reaction (no ALFA-tag on YfgM), both proteins were absent in the eluate. Complete blots in Supplementary Fig. 8. Color code used in sketch: PpiD (purple), YfgM (pink)

formation. Second, the peptide forms multiple specific contacts with the nanobody (Supplementary Table 3) along the whole length of the 13 amino acid ALFA core sequence (SRLEEELRRRLTE). In sum, nearly all residues of this sequence are involved in binding to NbALFA and/or stabilizing the α-helical peptide conformation. Due to the compact structure of the complex, the maximal displacement between an ALFA-tagged target and a fluorophore attached to NbALFA is well below 5 nm and can be reduced to < 3 nm by choosing adequate positions for fluorescent labeling. In order to minimize the potential influences of neighboring sequences on the ALFA-tag conformation, we placed the ALFA core sequence between prolines acting as insulators. Using this approach, the interaction of NbALFA with the ALFA-tag is largely independent of its localization within the tagged protein and is efficiently recognized when placed at either terminus of the POI or even in between two separately folded domains.

The strong interaction of NbALFA and the ALFA-tag is ideal for imaging applications, highly sensitive detection of target proteins, and purification of extremely low-abundant proteins from dilute lysates or under conditions where harsh washings with chaotropic agents are required. The slow dissociation, however, precludes a competitive elution of ALFA-tagged proteins under physiological conditions within a reasonable time frame. Based on the crystal structure of the NbALFA-ALFA peptide complex (Fig. 3), we site-specifically mutated the nanobody to increase its off-rate. A resin displaying the mutant nanobody (ALFA Selector[PE]) allows for native purifications of proteins and protein complexes from various lysates under physiological conditions by peptide elution (Figs. 4 and 5). ALFA Selector[PE] could even be applied for the selective enrichment of CD62L-positive lymphocytes from PBMC preparations (Fig. 6). We believe that this technique can easily be transferred to the highly validated recombinant Fab and scFv fragments that are

currently used for cell isolation approaches and similar purposes[39], or to other nanobodies recognizing surface markers that can easily be equipped with an ALFA-tag. Our technology can therefore contribute to current advances in biomedical research and therapy including the CAR-T technology[33].

The ALFA system stands out by its broad applicability. Using the ALFA system, a single transgenic cell line or organism harboring an ALFA-tagged target protein is sufficient for a wealth of different applications including (super-resolution) imaging, in vivo manipulation of proteins, in vitro detection by Western blot or even native pull-down applications aiming at detecting specific interaction partners or at isolating specific cell populations. The wide range of applications of the ALFA system provides the scientific community with a highly versatile tool, which will facilitate future scientific breakthroughs.

## Methods

**Non-cropped images.** Non-cropped versions of blots shown in Figs. 2a and 5c are included in Supplementary Information (Supplementary Figs. 4 and 8, respectively).

**Plasmids.** Plasmids used in this study are summarized in Supplementary Table 4.

**Non-commercial _E.coli_ strains.** MC4100 Δ*yfgM*[32] was used for the experiment described in Fig. 5c.

**Antibodies.** Antibodies used in this study are summarized in Supplementary Table 5.

**Protein expression and purification.** All recombinant proteins were expressed under the control of the Tac-promoter from expression vectors with ColE1 origin that confer resistance to Kanamycin.

The MBP fusion protein harboring multiple epitope tags, ALFA-shGFP2, TST-bdNEDD8-ALFA, NbCD62L-ALFA and non-tagged NbALFA were expressed as N-terminal $His_{14}$-bdSUMO fusions. For protein expression, E. coli was cultured in Terrific broth (TB) supplemented with 0.3 mM IPTG for 14–16 h at 23 °C. After harvest, E. coli cells were lysed in LS buffer (50 mM Tris/HCl pH7.5, 300 mM NaCl) supplemented with 15 mM imidazole/HCl pH7.5 and 10 mM DTT, and purified by binding to nickel-chelate beads. After extensive washing, proteins were eluted by on-column-cleavage with 100 nM bdSENP1 for 1 h at 4 °C[40,41]. ALFA-shGFP2-$His_6$ and $His_{14}$-bdSUMO-ALFA-shsfGFP were expressed and purified in a similar fashion; Elution was, however, performed using 250 mM Imidazole in buffer LS. For affinity determinations and binding studies from complex lysates, substrate proteins were additionally purified via size exclusion chromatography on a Superdex200 10/30 column (GE Healthcare). NbALFA harboring N- and C-terminal ectopic cysteines was expressed and purified as described for non-tagged NbALFA above. Labeling with maleimide-activated fluorophores was performed according to the manufacturer's manual.

**Immunizations.** All work involving animal experiments at NanoTag Biotechnologies complies with the relevant ethical regulations for animal testing and research. All experiments conducted do not require ethical approval, but are communicated to and accepted by the local authorities (LAVES Niedersachsen, Germany). Two alpacas were immunized 6 times with 0.5 mg ALFA peptide containing an additional N-terminal cysteine (CSRLEEELRRRLTE-Amide) conjugated to 1 mg MBS-activated KLH (0.9 mL + 0.9 mL adjuvant). The first immunization (day 0) was done using complete Freund's adjuvant, for all following immunizations (days 14, 28, 42, 56, and 112) incomplete Freund's adjuvant was used. In total 100 mL peripheral blood was taken from each animal 5 days after the last immunization and immediately incubated with 5000 IU mL$^{-1}$ heparin (Sigma) to prevent clotting.

**SdAb selection by affinity purification of B lymphocytes.** A total of 1 mL of T-Catch resin (IBA Lifesciences) was washed with B cell isolation buffer (PBS pH7.4, 1% BSA, 1 mM EDTA) and incubated with saturating amounts of a TST-bdNEDD8-ALFA fusion protein for 30 min rolling at RT. The resin was cleared from excess bait protein by extensively washing with B cell isolation buffer. In total 100 mL of blood sample was taken from alpaca immunized with ALFA peptide and immediately incubated with 5000 IU mL$^{-1}$ heparin (Sigma) to prevent clotting. From the fresh blood (<4 h past sampling) PBMCs were isolated using Ficoll-Paque PLUS (GE Healthcare). To remove residual serum, PBMCs were washed three times consecutively with B cell isolation buffer. PBMCs were passed three times over the antigen-loaded T-Catch resin before washing the resins with 10 CV B cell isolation buffer. Bound B cells were eluted from the resins by incubating with 2 μM bdNEDP1[40,41] for 30 min at RT. Total RNA (590 ng total) was extracted from the eluted B cells using NucleoSpin RNA plus kit (Machery Nagel) and used for a reverse transcription reaction (SuperScript IV Reverse Transcriptase; Thermo Fisher Scientific) with primer CaLl 02 (Supplementary Table 6). Nanobody-coding sequences were amplified by a two-step nested PCR using primers CaLl 01 and CaLl 02 (first

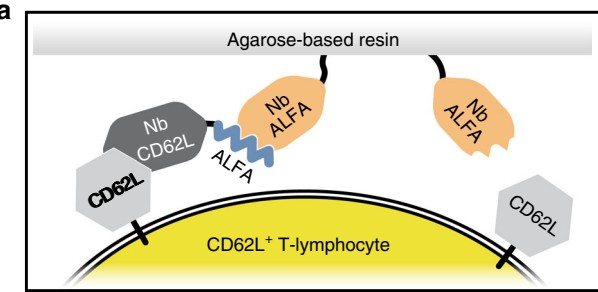

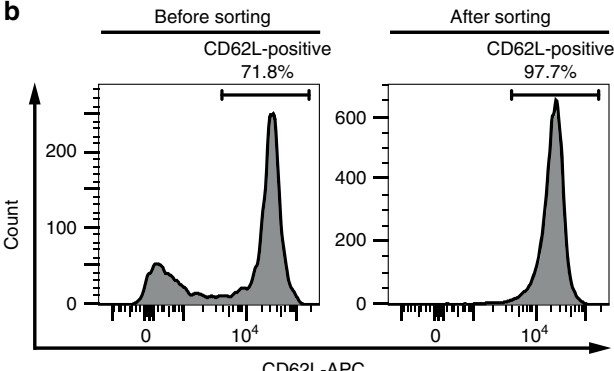

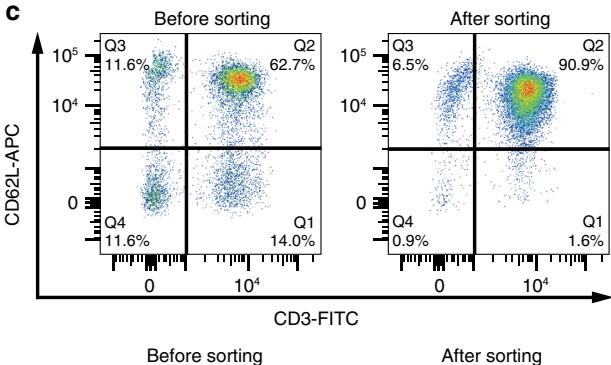

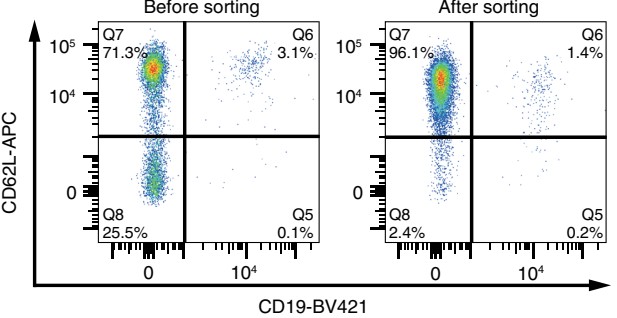

**Fig. 6** Isolation of naive lymphocytes using an ALFA-tagged nanobody recognizing CD62L. Total human PBMCs were left untreated (Before sorting) or isolated using an ALFA Selector$^{PE}$ resin loaded with an ALFA-tagged anti-human CD62L nanobody (After sorting). **a** A sketch of the affinity purification strategy. **b** Cells were stained with an anti-CD62L antibody and analyzed by flow cytometry. **c** The same cells as in **b** were stained with antibodies directed against CD3, CD19 and CD62L, and analyzed by flow cytometry. A forward scatter/side scatter gate was set on lymphocytes in all analyses (Supplementary Fig. 9). Color code used in **a**: NbALFA (orange), ALFA-tag (blue), CD62L (light gray), NbCD62L (dark gray), CD62L$^{+}$ T-lymphocyte (yellow)

PCR) and primers F1 and R1 (second PCR). The amplified sdAb library was directly cloned into a pQE80-derived prokaryotic expression vector in frame to an N-terminal His$_{14}$-bdSUMO-tag and a C-terminal FLAG-GFP-tag. In total 96 single clones were expressed in deep-well plates. After cell disruption, crude lysates were tested by ELISA for the presence of ALFA-reactive sdAbs. In total 16 clones showing signals > 5-fold above background were sequenced and further characterized biochemically. 13 of these clones could be grouped into two independent sequence families with 10 and 3 clones, respectively. NbALFA belonged to the most abundant family of clones that biochemically behaved most favorable.

**Crystallization and structure determination**. For crystal formation, non-tagged NbALFA was incubated with a 1.2-fold excess of ALFA peptide (Acetyl-PSRLEEELRRRLTEP-Amide; Genscript) for 60 min on ice. The complex was concentrated to ~25 mg mL$^{-1}$ and separated from the excess peptide by gel filtration on a Superdex75 10/30 column (GE Healthcare) equilibrated with 20 mM HEPES pH 7.4, 100 mM NaCl, 10% glycerol.

Crystals of the NbALFA-ALFA peptide complex were found in the F2 condition of the JCSG+ screen (Molecular Dimensions). This hit was further optimized using the Hampton Research Additive Screen. A crystal suitable for data collection was found in a drop containing 100 nL of precipitant solution (100 mM sodium citrate pH6.0, 2 M ammonium sulfate, 4% v/v tert-Butanol) and 100 nL of protein solution (10.1 mg mL$^{-1}$ NbALFA-ALFA peptide complex, 20 mM HEPES pH7.4, 100 mM NaCl, 10% glycerol) in a sitting-drop crystallization plate. X-ray diffraction data collection was performed at BL14.2 at the BESSY II electron storage ring operated by the Helmholtz-Zentrum Berlin, Germany[42]. The data were processed using DIALS[43], molecular replacement was performed with Phaser[44] using the structure with PDB code 5VNV as an atomic model. The structure was built using Phenix Autobuild[45] and Coot[46]. The structure was refined using Refmac5[47].

**Affinity determinations**. All surface plasmon resonance (SPR) binding experiments were performed on a Reichert SR 7500DC benchtop instrument at 20 °C and a flow rate of 40 μL min$^{-1}$ on HC1000m SPR sensor chips (Xantec Biotechnologies) in PBS buffer containing 0.002% (v/v) Tween-20, pH 7.4. Data were analyzed using TraceDrawer v1.7.1. The HC1000m sensorchip was activated with 1-ethyl-3–3(3-dimethylaminiopropyl)-carbodiimide and N-hydroxysuccinimide (EDC-NHS) according to the manufacturers instructions. The ligand was pre-diluted to 5 μM in concentration buffer (50 mM sodium acetate pH 5.0) and was injected over the activated chip surface at a flow rate of 15 μL min$^{-1}$ to a finalresponse of 150 μRIU (TST-NEDD8-ALFA) and 650 μRIU (GFP-ALFA) (RIU: refractive index units; response recorded). The chip surface was subsequently inactivated with 1 M ethanolamine pH 8.5. The chips with immobilized ligands were equilibrated with PBS pH 7.4 containing 0.002% (v/v) Tween-20 (PBS-T). The analytes (NbALFA$^{PE}$, or NbALFA) were serially diluted in PBS-T and injected over the chip surface. Association was followed for 4.5 min and dissociation was followed for 15 min for NbALFA$^{PE}$ and accordingly 45 min for NbALFA. For each analyte, two buffer injections were performed as a reference (buffer reference). The recorded binding was double-referenced against an empty reference channel recorded in parallel and the buffer reference. The surface was regenerated after each analyte injection by injecting 100 mM Glycine pH 2.2, 150 mM NaCl for 2.5 min. To accurately determine the dissociation constant of NbALFA, the decrease in response was followed for > 7 h.

**Transfection of mammalian cells**. For immunofluorescence experiments, 3T3 (DSMZ no. ACC 173), COS-7 (DSMZ no. ACC 60) or HeLa (DSMZ no. ACC 57) cells were transiently transfected with 0.2–1 μg of plasmids listed in Supplementary Table 4, using the PolyJet transfection kit (SignaGen) or lipofectamine 3000 (Thermo Fisher Scientific, cat. no. L3000–001) according to the manufacturer's recommendations. 3T3 and COS-7 cells were seeded on 12-well plates and HeLa in 8-well chambered coverglass (ibidi, cat. no. 80827). Cells were incubated with transfection reagents for 24–48 h. For co-expression experiments, plasmid DNA was premixed in a 1:1 ratio and further processed as described above.

**Microscopy setups**. Epifluorescence images were obtained with an Axio Observer Z1, (Carl Zeiss GmbH) equipped with a ×20 dry lens or a ×40 oil immersion lens. Confocal and STED microscopy were acquired using an inverse 4-channel Expert Line easy3D STED setup (Abberior Instruments GmbH, Göttingen, Germany). The setup was based on an Olympus IX83 microscope body equipped with a plan apochromat ×100 1.4 NA oil-immersion objective (Olympus). In total 640 nm excitation laser (Abberior Instruments GmbH) pulsed at 40 MHz was used. Stimulated depletion was achieved with a 775 nm STED laser (Abberior Instruments GmbH) pulsed at 40 MHz with an output power of ~1.250 W. Fluorescence signal was detected using APD detectors (Abberior Instruments GmbH). The operation of the setup and the recording of images were performed with the Imspector software, version 0.14 (Abberior Instruments GmbH). DNA-PAINT imaging was carried out on an inverted Nikon Eclipse Ti microscope (Nikon Instruments) with the Perfect Focus System, applying an objective-type TIRF configuration with an oil-immersion objective (Apo SR TIRF ×100, NA 1.49, Oil). Two lasers were used for excitation: 561 nm (200 mW, Coherent Sapphire) or 488 nm (200 mW, Toptica iBeam smart). The laser beam was passed through a cleanup filter (ZET488/10x or

ZET561/10x, Chroma Technology) and coupled into the microscope objective using a beam splitter (ZT488rdc or ZT561rdc, Chroma Technology). Fluorescence light was spectrally filtered with two emission filters (ET525/50 m and ET500lp for 488 nm excitation and ET600/50 and ET575lp for 561 nm excitation, Chroma Technology) and imaged on a sCMOS camera (Andor Zyla 4.2) without further magnification, resulting in an effective pixel size of 130 nm after 2 × 2 binning. Camera Readout Sensitivity was set to 16-bit, Readout Bandwidth to 540 MHz.

**Epifluorescence and STED microscopy imaging**. Transiently transfected 3T3 or COS-7 cells were fixed in either 4% paraformaldehyde (PFA) (w/v) or 2% glutaraldehyde (GA) (v/v) for 30 min at room temperature (RT). Alternatively, fixation was performed in ice cold methanol for 15 min at −20 °C. Cells were blocked and permeabilized in PBS containing 10% normal goat serum (v/v) and 0.1% Triton-X 100 (v/v) for 15 min at RT. Fluorescently labeled NbALFA (FluoTag®-X2 anti-ALFA AbberiorStar635P, NanoTag Biotechnologies #N1502-Ab635P-L) was diluted 1:500 in PBS containing 3% normal goat serum and 0.1% Triton-X 100 (v/v). The cells were incubated in this staining solution for 1 h at RT and subsequently washed 3 times for 5 min with PBS. To stain the nucleus, DAPI (0.4 μg mL$^{-1}$) was included in one of the PBS washing steps. Cover slips were mounted on glass-slides using Mowiol solution, dried at 37 °C and imaged. Constructs expressed at the cell-surface were co-stained with anti-FLAG® M2 (primary antibody, Sigma, F1804) and FluoTag®-X2 anti-mouse IgG Atto488 (secondary nanobody, NanoTag Biotechnologies #N1202-At488-L) diluted 1:1000 and 1:500 respectively, in PBS containing 3% normal goat serum and 0.1% Triton-X 100 (v/v).

**DNA-labeling of NbALFA for DNA-PAINT and data analysis**. NbALFA was first coupled to a single stranded DNA as described before[48]. In brief, NbALFA bearing ectopic N- and C-terminal cysteines was reduced on ice with 5 mM of tris(2-carboxyethyl)phosphine (TCEP, Sigma-Aldrich, #C4706) for 2 h. After removing excess TCEP using 0.5 mL Amicon Ultra spin filters with a molecular weight cut-off (MWCO) of 10 kDa (Merck, #UFC500324), the nanobody was conjugated with a 50-fold molar excess of maleimide-DBCO crosslinker (Sigma-Aldrich, #760668) at 4 °C overnight. Using a 10 kDa MWCO Amicon spin filter, the excess of crosslinker was removed and the nanobody was further incubated with 10 fold molar excess of a single stranded DNA containing an azide group on its 5′-end and an Atto488 fluorophore on its 3′ end (P3 docking strand: 5′-Azide-TTTCTTCATTA-Atto488–3′) obtained from Biomers.net GmbH) for 2 h at room temperature. Excess of DNA was removed by using a size exclusion chromatography column (Superdex® Increase 75, GE Healthcare) on an Äkta pure 25 system (GE Healthcare).

For staining, cells were prefixed with 0.4% (v/v) Glutaraldehyde and 0.25% (v/v) Triton X-100 in PBS at pH7.2 for 90 s. The main fixation was performed with 3% glutaraldehyde in PBS for 15 min. Afterwards the sample was reduced with 1 mg mL$^{-1}$ sodium borohydride (Carl Roth, #4051.1) for 7 min and washed 4× (1x fast, 3 × 5 min) with PBS. Blocking and additional permeabilization was performed for 90 min in 3% (w/v) BSA + 0.2% (v/v) Triton X-100 in PBS. Next, NbALFA conjugated to the P3 docking strand was added to the sample to a final concentration of 5 μg mL$^{-1}$ in dilution buffer (3% w/v BSA in PBS) and incubated for 1 h at RT. The sample was washed three times for 5 min in PBS and then incubated for 10 min with 1:5 diluted fiducial markers, 90-nm gold particles (Cytodiagnostics, #G-90–100), residual gold was washed away with PBS. Cells were kept at 4 °C until they were used for imaging.

For DNA-PAINT imaging, transfected cells were screened for a certain phenotype with 488 nm laser excitation at 0.01 kW cm$^{-2}$. After acquisition of the 488 nm channel, the excitation was switched to 561 nm, focal plane and TIRF angle were readjusted and imaging was subsequently performed using ~2.5 kW cm$^{-2}$ 561 nm laser excitation. P3-imager strand (5′-GTAATGAAGA-Cy3b-3′) concentration was chosen to minimize double-binding events. ALFA-tag imaging was performed using an imager concentration of 1 nM P3-Cy3b in Buffer C (PBS + 500 mM NaCl). All imaging was performed in 1×PCA (Sigma-Aldrich, #37580)/1×PCD (Sigma-Aldrich, #P8279)/1×Trolox (Sigma-Aldrich, #238813) in Buffer C and cells were imaged for 40,000 frames at 100 ms camera exposure time. 3D imaging was performed using a cylindrical lens in the detection path[49]. Raw data movies were reconstructed with the Picasso software suite[26]. Drift correction was performed with a redundant cross-correlation and gold particles as fiducials. Final images obtained had a localization precision of < 10 nm calculated via a nearest neighbor analysis[50]. Rendering was performed via the recently published SMAP software.

**Impact of ALFA-tags on the localization of EGFP**. Transiently transfected 3T3 cells were imaged using an epifluorescence microscope (Axio, Zeiss) equipped with a ×40 1.3 NA oil lens. For cells transfected with either pCMV ALFA-EGFP, pCMV EGFP-ALFA, or pEGFP-N1, 107–133 cells were imaged in a total of six to seven individual images. For each individual image, cells were grouped and counted according to the localization of EGFP (slightly nuclear, equally distributed, other). The fraction of cells in each group was statistically analyzed using Student's t-test.

**Western Blots with COS-7 lysates**. Transfected cells from a confluent 10 cm petri dish were washed with PBS and lysed in 2 mL SDS sample buffer. Lysates were resolved by SDS-PAGE and transferred to a nitrocellulose membrane. After

blocking with 5% milk powder in TBS-T, membranes were incubated with mouse anti-tubulin (Synaptic Systems #302 211; 1:1000 dilution) followed by a FluoTag®-X2 anti-Mouse IgG IRDye680 (NanoTag Biotechnologies #N1202; 1:1000 dilution) and FluoTag®-X2 anti-ALFA IRDye800 (NanoTag Biotechnologies #N1502; 1:1000 dilution). Membranes were scanned using Odyssey CLx (Li-COR).

**Dotblot assay.** A serial dilution of MBP fused to FLAG®-, HA-, myc- and ALFA-tags was prepared in PBS pH7.4, 0.1 µg mL⁻¹ BSA. In total 1 µL of each dilution was spotted on nitrocellulose membranes. Membranes were blocked and washed with 5% milk powder in TBS-T. Established monoclonal antibodies (anti-FLAG® M2, Sigma #F1804; anti-myc 9E10 Synaptic Systems #343 011; anti-HA F-7, SantaCruz #sc-7392) were used in combination with a secondary goat anti-Mouse IgG IRDye800CW (Li-COR #925–32210, dilution 1:500) to detect FLAG®-, myc- and HA-tag, respectively. The ALFA-tag was detected using a FluoTag®-X2 anti-ALFA nanobody (NanoTag Biotechnologies #N1502) directly coupled to IRDye800CW. All primary antibodies and the nanobody were used at 2.7 nM final concentration. Detection of MBP by a rabbit polyclonal serum recognizing MBP (Synaptic Systems, dilution 1:500) and an anti-rabbit IgG IRDye680RD (Li-COR #925–68071, dilution 1:5000) served as an internal loading control. Membranes were scanned using Odyssey CLx (Li-COR). Quantifications were performed using ImageStudioLight (Li-COR).

**Off-rate assays.** In total, 20 µL ALFA Selector^ST or ALFA Selector^PE (NanoTag Biotechnologies #1511 and #1510) was saturated with the respective recombinant target protein. After washing 4× with PBS, the beads were suspended in a 10-fold excess of PBS containing 200 µM free ALFA peptide and mixed at 25 °C. Control reactions were carried out without peptide. At indicated time points, specific elution from the beads was quantified using the GFP fluorescence released into the supernatant (Q-Bit 3.0; Thermo-Fischer Scientific). Three independent experiments were performed in parallel. Mean values, standard deviations and exponential fits were calculated using GraphPad Prism 5.0. Photographic pictures were taken upon UV illumination using a Nikon D700 camera equipped with a 105 mm macro lens (Nikon).

**Resistance towards stringent washing and pH.** In total, 15 µL of ALFA Selector^ST or ALFA Selector^PE saturated with either ALFA-shGFP2 or shGFP2-ALFA were washed with PBS and incubated with 100 µL of the indicated substances for 60 min at RT. Photos were taken after sedimentation of the beads upon UV illumination. To assay for pH resistance, the same beads were incubated with 150 mM NaCl buffered to various pH (100 mM Glycin-HCl, pH2.2; 100 mM Na-Acetate pH4.5, 100 mM Tris–HCl pH7.5, 100 mM Carbonate pH9.5) for 30 min at RT. The resin was washed twice with the same buffer. Photos were taken after equilibrating with PBS.

**Affinity purifications from *E. coli* and HeLa lysates.** To obtain defined input materials for pull-down experiments from *E. coli* or HeLa lysates, respective mock lysates were blended with 3 µM of purified ALFA-tagged shGFP2. In total 1 mL of each lysate/substrate mixture was incubated with 25 µL of ALFA Selector^ST or ALFA Selector^PE for 1 h at 4 °C. An analogous resin without immobilized sdAb (Selector Control, NanoTag Biotechnologies #N0010) served as a specificity control. After washing 3 times with 600 µL of PBS, the resins were transferred into MiniSpin columns (NanoTag Biotechnologies #A1001). Excess buffer was removed by centrifugation (3000 × g, 30 s) before incubating twice for 10 min at RT with 50 µL each of 200 µM ALFA peptide in PBS. Proteins remaining on the beads were afterwards eluted with SDS sample buffer. In total 0.5 µL (*E. coli*) or 1.5 µL (HeLa) of input and non-bound fractions were resolved by SDS–PAGE (12%) and Coomassie staining. Shown eluate fractions correspond to the material eluted from 1 µL of the respective resins.

**Native pull-down of the *E.coli* YfgM–PpiD complex.** A *yfgM* deletion strain was complemented with either C-terminally ALFA-tagged or untagged YfgM expressed from a pSC-based low-copy vector under control of the endogenous promoter. Membrane protein complexes were solubilized from total lysates prepared in buffer LS (50 mM Tris pH7.5, 300 mM NaCl, 5 mM MgCl₂) using 1% DDM for 1 h on ice[51]. Both lysates were incubated with 20 µL of ALFA Selector^PE resin for 1 h at 4 °C on a roller drum. After washing in PBS containing 0.3% DDM, bound proteins were eluted under native conditions by sequentially incubating twice with 50 µL PBS containing 200 µM ALFA peptide at RT. Samples corresponding to 1/800 of the input and non-bound material or 1/80 of eluate fractions were resolved by SDS–PAGE. Analysis was performed by Western blotting using a polyclonal rabbit serum raised against the YfgM-PpiD complex[32] (dilution 1:500) followed by an HRP-conjugated goat anti-rabbit IgG (Dianova, dilution 1:10,000). Blots were developed using the Western Lightning Plus-ECL Kit (Perkin Elmer) and imaged using a LAS 4000 mini luminescence imager (Fuji Film).

**Preparation of human PBMCs.** Human blood was obtained from a healthy volunteer participant. The experiment was approved by the ethical review committee of the University Medical Center Göttingen (case number 11/6/17).

Peripheral blood mononuclear cells (PBMCs) were obtained from fresh blood using standard density gradient centrifugation. Briefly, 60 mL of fresh blood was diluted with 40 mL of phosphate-buffered saline (PBS) supplemented with 1 mM EDTA and placed on top of a layer of CELLPURE Roti-Sep 1077 (Carl Roth) in 50 mL LEUCOSEP tubes (Greiner Bio-One) and centrifuged at 800 × g for 20 min at room temperature. Subsequently, the PBMC-containing layer was collected and washed five times in cold PBS + EDTA to remove platelets.

**Isolation of CD62L-positive lymphocytes.** Approximately 2 × 10⁷ PBMCs were passed by gravity flow through an ALFA Selector^PE resin loaded with an ALFA-tagged anti-human CD62L nanobody, followed by extensive washing with PBS supplemented with 1 mM EDTA and 1% (w/v) bovine serum albumin. Subsequently, bound cells were eluted in the same buffer containing 200 µM ALFA peptide.

**Flow cytometry.** For flow cytometric analysis, roughly 1 × 10⁶ cells per sample were stained with monoclonal antibodies to human CD3 (clone UCHT1) coupled to FITC (BioLegend #300405, dilution 1:250), CD19 (clone HIB19) coupled to BV421 (BioLegend #302233, dilution 1:50) and CD62L (clone DREG-56) coupled to APC (BD Biosciences #561916, dilution 1:12.5). After washing with PBS supplemented with EDTA and BSA, cells were measured using an LSRII cytometer (BD Biosciences). Data were analyzed using FlowJo (v10) software (FlowJo, LLC, Ashland, OR, USA). The gating strategy is illustrated in Supplementary Fig. 9.

**Reporting summary.** Further information on research design is available in the Nature Research Reporting Summary linked to this article.

## Data availability
The atomic coordinates and structure factors have been deposited in the Protein Data Bank (www.rcsb.org/structure/6I2G, code 6I2G). Primary data of graphs shown in Fig. 4b, and Supplementary Figs. 2, 6 and 7 are available in the Source Data file. All other datasets generated during and/or analyzed during the current study are available from the corresponding authors (F.O. and S.F.) on reasonable request.

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

## Acknowledgements

We thank Alexandra Lück and Verena Pape for excellent technical assistance, the scientists at BM14.2, HZB, Berlin for their support during X-ray diffraction data collection, Jonas Ries for SMAP software support and Florian Schueder, Alexander Auer, and Maximilian T. Strauss for support with DNA-PAINT experiments. We further thank Mihaela Serpe for directing our attention to potential in vivo applications, Blanche Schwappach for her generous support regarding affinity determinations, and Eugenio F. Fornasiero and Silvio Rizzoli for comments on the manuscript. F.O. and S.S.-I. were supported by the DFG through Cluster of Excellence Nanoscale Microscopy and Molecular Physiology of the Brain (CNMPB). This work was partially supported by grants from the Swedish Research Council (2014–5667) and the Swedish Cancer Society to P.S. T.S. acknowledges support from the DFG through the Graduate School of Quantitative Biosciences Munich (QBM). R.J. is supported by the DFG through the Emmy Noether Program (DFG JU 2957/1–1), the SFB 1032 (Project A11) and by the ERC Starting Grant (MolMap; 680241), by the Max Planck Society and the Center for Nanoscience (CeNS).

## Author contributions

S.F. and F.O. conceived the project. S.F., H.G., and F.O. were involved in experimental design, nanobody selection and development of the ALFA system. S.F., H.G., M.K., and F.O. performed and analyzed most of the experiments. A.R. initially characterized the NbALFA$^{PE}$ mutant. M.M.-C. and P.S. solved and analyzed the NbALFA-ALFA peptide structure. N.E. helped with the cell isolations and performed the FACS analysis. S.S.-I. contributed critical reagents for DNA-PAINT and affinity measurements. T.S. and R.J. performed and analyzed DNA-PAINT experiments. S.F., H.G., M.K. and F.O. wrote the manuscript. All authors read and commented on the manuscript.

## Additional information

**Competing interests:** S.F., H.G., F.O., M.M.-C., and P.S. are inventors on a pending European patent application covering the ALFA system and its use. S.F., H.G., and F.O. are shareholders of NanoTag Biotechnologies GmbH. The remaining authors declare no competing interests.

