## [Peer Review File · Nature Communications]

Reviewers' comments:

Reviewer #1 (Remarks to the Author):

In this paper, authors report on the identification of a nanobody targeting specifically to a short structured (alpha helical) oligopeptide, named ALFA. The interaction seems to be strong enough to resist various chemicals and thus the nanobody can be employed for pull downs, for imaging on fixed samples, in super resolution imaging. The structure of the complex is determined by crystallography and a nanobody variant is generated with a faster off rate, which might be useful to purify ALFA tagged highly fragile (membrane) complexes.

Similar affinity reagents to oligopeptide tags are already in use, however, as the authors claim, most of these tag-binder systems cannot be used for all of the above mentioned applications. However, here as well there is a need to switch from one nanobody to its variant depending on the application.

To show the potential of the Nb-ALFA the authors use sometimes smart set-ups. (using GFP-ALFA tagged proteins) Also the isolation of the Nb-ALFA is ingenious and non-conventional.

Despite this, there is a clear lack of novelty or groundbreaking technology: Nanobodies have been used multiple times in super resolution microscopy, for pull down experiments, even in western blot or dot blots, or for cell sub-type isolation via FACS.

Major comment:

The affinity of the Nb-ALFA for an ALFA-tagged target should be measured, preferably by Biacore or any other biosensor instead of being inferred from competition with free peptide. It should not be too difficult to coat a biosensor chip with a ELFA-tagged protein and to flow different concentrations of the Nb_ALFA over it to measure the k_{on} and k_{off} rates.

Minor comment:

The authors claim that the Nb-ALFA is working in western blot. This might well be, however, they only show that they work in dot blot. It is also written that Nanobodies are usually not working in western blot. This is not entirely true, many Nbs work in western blot on condition that the target separated in SDS-PAGE was not reduced.

Reviewer #2 (Remarks to the Author):

In this paper, Goetzke et al report the use of a previously published peptide as epitope that can be recognized by a newly selected nanobody. The authors provide a convincing and thorough characterization of the properties of this interaction pair and demonstrate several important applications. The data is of high quality and, while reading, I became very enthusiastic about the work and its importance. Nonetheless, I cannot recommend publication in its current form because it lacks crucial information needed to reproduce the work. Without this information, it feels more like an advertisement for a new product than a scientific publication.

1. ALFA sequence: about the choice for the ALFA sequence, the authors write: The ALFA-tag sequence (Fig.1a) originates from an artificial peptide reported to form a stable α -helix in solution. Here the authors cite Petukhov, J. Pept. Sci 2009 (ref 23). However, the exact sequence used in the current study (SRLEEEELRRRLTE) cannot be found in that earlier work. The earlier paper introduces 80 different sequences of which only one is very similar to the one reported here (SRLEEEELRRRLGGY). The authors do not provide any justification for the choice of this sequence, nor for the differences they introduced.

2. Nanobody production: While the whole paper revolves around the selection and characterization of new nanobodies against this previously published peptide, the methods give no details about the constructs nor the purification and conjugation protocols. Instead, the new antibodies developed in this work are listed as the first three entries in Table M2 as being obtained from NanoTag Biotechnologies, the employer of the lead authors. This could perhaps be justifiable if the authors do not wish to disclose these things for commercial reasons. However, in that case they

should produce a manuscript in which these tools are used for some biological discovery and not one that is completely about these tools.

3. Sequence information: while the authors nicely report the sequence of their SuperTight nanobody, I could not find this information for the PeptideElution variant. Instead the authors write: To allow for an efficient competitive elution of ALFA-tagged target proteins, we intended to weaken NbALFA. We identified NbALFAPE (for Peptide Elution) following a rational approach based on the NbALFA-ALFA peptide structure. In my opinion more information about the approach and the outcome is needed to justify publication in a scientific journal.

4. DNA-PAINT experiments: The methods (p22, l30) suggest that conjugation of the docking strand to the nanobody was performed after fixation and permeabilization. Is this really how the experiment was done? If so, why is that? Is the product very unstable? In addition, details about the conjugation are missing. What chemistry was used for this?

Reviewer #1 (Remarks to the Author):

In this paper, authors report on the identification of a nanobody targeting specifically to a short structured (alpha helical) oligopeptide, named ALFA. The interaction seems to be strong enough to resist various chemicals and thus the nanobody can be employed for pull downs, for imaging on fixed samples, in super resolution imaging. The structure of the complex is determined by crystallography and a nanobody variant is generated with a faster off rate, which might be useful to purify ALFA tagged highly fragile (membrane) complexes.

Similar affinity reagents to oligopeptide tags are already in use, however, as the authors claim, most of these tag-binder systems cannot be used for all of the above mentioned applications. However, here as well there is a need to switch from one nanobody to its variant depending on the application.

This is true. Our point, however, is that the ALFA-tag is extremely versatile. For the scientist at the bench the largest effort is to produce a multitude of plasmids, cell-lines or even transgenic animals harboring adequate tags fused to their protein(s) of interest. Our goal was to create a single tag that can be used for all possible applications. This idea is detailed in our introduction and explicitly in the last paragraph of the discussion:

"Using the ALFA system, a single transgenic cell line or organism harboring an ALFA-tagged target protein is sufficient for a wealth of different applications including (super-resolution) imaging, in-vivo manipulation of proteins, in-vitro detection by Western blot or even native pull-down applications aiming at detecting specific interaction partners or at isolating specific cell populations."

To show the potential of the Nb-ALFA the authors use sometimes smart set-ups. (using GFP-ALFA tagged proteins) Also the isolation of the Nb-ALFA is ingenious and non-conventional.

Despite this, there is a clear lack of novelty or groundbreaking technology: Nanobodies have been used multiple times in super resolution microscopy, for pull down experiments, even in western blot or dot blots, or for cell sub-type isolation via FACS.

Indeed, it is true that there are tags for each specific application. We explicitly discuss this fact e.g. in our introduction and in Table S1. As detailed above, our idea was to create a tag that can serve all of these applications with the best possible performance at the same time. Such a tag is, to our knowledge, not available so far (see, e.g. Table S1).

Major comment:

The affinity of the Nb-ALFA for an ALFA-tagged target should be measured, preferably by Biacore or any other biosensor instead of being inferred from competition with free peptide. It should not be too difficult to coat a biosensor chip with a ELFA-tagged protein and to flow different concentrations of the Nb_ALFA over it to measure the k_{on} and k_{off} rates.

We thank the reviewer for this suggestion. We now obtained affinity data by surface plasmon resonance (SPR) binding assays, which we included as Figure S5 in the manuscript. We also tried microscale thermophoresis as an alternative method. Here, however, the affinity of the NbALFA:GFP-ALFA interaction was too strong to be accurately determined by this method. The now included new information made the manuscript stronger, since we can now mention that the high affinity nanobody binds with a K_d of ~26 pM and the peptide elutable nanobody (NbALFA^{PE}) with a K_d of ~11 nM to the ALFA-tagged proteins.

The new information is now incorporated in the revised manuscript at several positions: Abstract (p2, line8) and Results (p10) and we included a new Supplementary Figure (Fig.S5) showing the SPR data. The K_d value obtained for NbALFA is now also mentioned in Table S1. The methods used for affinity determination are now described in a new methods section on p. 22.

Minor comment:

The authors claim that the Nb-ALFA is working in western blot. This might well be, however, they only show that they work in dot blot. It is also written that Nanobodies are usually not working in western blot. This is not entirely true, many Nbs work in western blot on condition that the target separated in SDS-PAGE was not reduced.

We show "real" Western blot data in Fig. 2A and Fig. S4A using the NbALFA. Here, as described in the main text, the figure legend and the methods section, lysates of COS-7 cells transfected with constructs encoding a N-terminally ALFA-tagged vimentin were resolved by (reducing) SDS-PAGE. After blotting, ALFA-tagged vimentin was detected using NbALFA directly coupled to IRDye800CW.

There are rather few examples described of nanobodies that detect their targets on Western blot membranes after SDS-denaturation and reduction (i.e. Itoh et al. Plos One (2014), Braun et al. Sci Rep. (2015), Maidorn et al. MAbs (2019)). Therefore, we carefully wrote "Nanobodies *often* fail to detect proteins in WBs". Our own experience match with the recurrent idea that most nanobodies fail to bind to SDS-denatured and reduced targets in Western blot applications, because the three-dimensional epitope recognized by the nanobody typically cannot refold after removal of SDS. The fact that nanobodies prefer structural epitopes (instead of linear SDS-denatured targets) is commonly explained by their lack of a light chain and the unconventionally long CDR3, which provide nanobodies with the natural tendency to prefer structural epitopes (for example see Dmitrev et al. JBC (2016)). We are not aware of any publication showing that nanobodies have a preference to bind to non-reduced, but SDS-denatured targets.

Reviewer #2 (Remarks to the Author):

In this paper, Goetzke et al report the use of a previously published peptide as epitope that can be recognized by a newly selected nanobody. The authors provide a convincing and thorough characterization of the properties of this interaction pair and demonstrate several important applications. The data is of high quality and, while reading, I became very enthusiastic about the work and its importance.

Thank you!

Nonetheless, I cannot recommend publication in its current form because it lacks crucial information needed to reproduce the work. Without this information, it feels more like an advertisement for a new product than a scientific publication.

We can only in part follow this perception.

1) This manuscript indeed details on the sound characterization of a novel epitope tag. The advantage of such study is that all the properties and potential applications of such tag are thoroughly evaluated and set into a common frame (very different than many commercial tools used commonly in research without any available validation data).

We strongly believe that the development of novel techniques in science is of great relevance and may have great impact. As a matter of fact, a large number of Nobel prizes have been awarded for “technical” advances. This is, of course, mentioned knowing that the impact of our technology is not worth a Nobel price...

2) We also believe that – besides the description of a novel tag system – our paper has a meta-level that goes beyond this "simple" technical aspect: It describes a new approach allowing to develop affinity tags (and the respective binders) with pre-defined properties. The main point here is that the sequence of the tag is chosen using rational considerations, which allows to pre-determine its properties. This is in contrast to most other epitope tags, which have been found as by-products during the development of antibodies.

3) Finally, we are also providing detailed method section where every experiment is reasonably explained. The descriptions given should allow to fully reproduce and validate our experiments.

1. ALFA sequence: about the choice for the ALFA sequence, the authors write: The ALFA-tag sequence (Fig.1a) originates from an artificial peptide reported to form a stable α -helix in solution. Here the authors cite Petukhov, J. Pept. Sci 2009 (ref 23). However, the exact sequence used in the current study (SRLEEELRRRLTE) cannot be found in that earlier work. The earlier paper introduces 80 different sequences of which only one is very similar to the one reported here (SRLEEELRRRLGGY). The authors do not provide any justification for the choice of this sequence, nor for the differences they introduced.

We would like to split this issue into two separate points:

a) Selection criteria

Reviewer #2 cites the first sentence of our results. *"The ALFA-tag sequence (Fig.1a) originates from an artificial peptide reported to form a stable α -helix in solution"*. Our selection criteria are further detailed in the following sentence: *"It was selected based on the following properties: i) The sequence is absent in common eukaryotic model systems, ii) it is hydrophilic and neutral at physiological pH and iii) it does not contain any primary amines that are modified by aldehyde-containing fixatives."*

Most of the sequences given in ref 23 actually fail to fulfil these criteria, mostly be-

cause they are exceedingly hydrophobic or contain lysines. As our purpose was not to analyze all of sequences given exhaustively, but to start with the development of one tag/binder system, we pragmatically selected two remaining peptides based on two additional criteria: i) "Good" balance between hydrophilic and hydrophobic amino acids and ii) high helix propensity. From these two sequences, the ALFA tag was the only one giving a good immune response (this criterion was unfortunately set by the immunized animals...).

b) Deviations between the ALFA sequence and the respective Petukov sequence:

We do not believe that it is essential for the described approach to fully understand the origin of the final ALFA tag sequence, but we felt it was important and fair to acknowledge that we did not design the sequence entirely *de novo* but built on an already existing peptide sequence with (partially) known properties. The alpha-helical peptides described in Petukhov et al. are all followed by the common tri-peptide GGY. This tri-peptide is, however, irrelevant for the stability of the ALFA helix. This fact is mentioned in legend to table 2 of the respective publication and the corresponding methods section. In fact, the Gly-Gly spacer is used in order NOT to disturb with the secondary structure of the preceding peptide sequence while the C-terminal Tyr is only introduced to allow for a better spectroscopic detection. The core sequence (**SRLEELRRRL**) is therefore presumably sufficient to form a reasonably stable alpha-helix in solution.

The ALFA core sequence (**SRLEELRRRLTE**) indeed contains two additional C-terminal amino acids (TE). Those were introduced i) as a consequence of the applied cloning technique and ii) to render the peptide neutral at physiological pH (i.e. to be fully in accordance with our selection criteria – see above).

However, we still believe these technicalities are not fully relevant for the final function of the ALFA tag and the nanobody binding to it.

To address the two separate sub-points raised by reviewer#2 we now modified the paragraph in question (p5). It now reads:

*The ALFA-tag sequence (Fig.1a) is **inspired** by an artificial peptide reported to form a stable α -helix in solution²³. It was selected based on the following properties: **i) It features a high alpha-helical content** ii) the sequence is absent in common eukaryotic model systems, iii) it is hydrophilic and neutral at physiological pH **while retaining moderate hydrophobic surfaces** and vi) it does not contain any primary amines that are modified by aldehyde-containing fixatives."*

2. Nanobody production: While the whole paper revolves around the selection and characterization of new nanobodies against this previously published peptide, the methods give no details about the constructs nor the purification and conjugation protocols. Instead, the new antibodies developed in this work are listed as the first three entries in Table M2 as being obtained from NanoTag Biotechnologies, the employer of the lead authors. This could perhaps be justifiable if the authors do not wish to disclose these things for commercial reasons. However, in that case they should produce a manuscript in which these tools are used for some biological discovery and not one that is completely about these tools.

As mentioned above: We believe the impact of the current manuscript goes far beyond the "simple" description of a new epitope tag (and I guess this is, why the reviewer himself "became very enthusiastic about the work and its importance").

Regarding nanobody production and availability of reagents we in part understand the concerns of reviewer#2, however, even in the very top journals with extremely high impact it is quite common (even explicitly allowed) to describe new methods without making all sequences, techniques and details immediately available. This is accepted also because especially companies have to protect their intellectual properties in order to survive (which actually is the main conceptual difference to "academic" science).

On the other side, it is indeed the responsibility of the authors to make the tools *as available as possible* in order to allow other researchers to reproduce the data shown. However, even in high impact manuscripts, self-build microscopes or other complex devices are described, that only a couple of lab might be able to assemble in order to reproduce the data, which does not mean that the data is wrong or of lower quality.

We guaranty reproducibility by explicitly citing the used products (see Table M2) and its suppliers (indeed: sometimes NanoTag). This approach is not uncommon – just an example: Every researcher using an M2 anti-FLAG antibody would (if at all) simply cite "Sigma-Aldrich #F-3156" without Sigma providing the exact sequence of the M2 antibody or giving away the M2 cell line. We, however, *in addition* give an almost complete description of how these reagents can be produced.

- a) We give the full sequence of the high-affinity ALFA binder, i.e. everyone can freely use this sequence for academic purposes, be it for recombinant expression or intracellular applications, using appropriate expression systems (as detailed in the methods section).
- b) In the first paragraph of the results we explicitly state that NbALFA has been labeled via ectopic cysteines and give a reference (Ref. 24, Pleiner et al.) describing in detail the applied method for site-specific labeling of nanobodies via ectopic cysteines.
- c) We cite the number of our pending patent application.

Altogether, any researcher "skilled in the art" will be able to reproduce the experiments – be it with reagents purchased from NanoTag or with similar reagents made in his/her lab.

In order to summarize the procedure for production of fluorescently labeled NbALFA, we now expanded the methods section with a compilation of the already existing descriptions regarding the expression, purification and labeling of NbALFA as follows (p.20):

"NbALFA harboring N- and C-terminal ectopic cysteines was expressed and purified as described for non-tagged NbALFA above. Labeling with maleimide-activated fluorophores was performed as previously described elsewhere²⁴."

3. Sequence information: while the authors nicely report the sequence of their SuperTight nanobody, I could not find this information for the PeptideElution variant. Instead the authors write: To allow for an efficient competitive elution of ALFA-tagged target proteins, we intended to weaken NbALFA. We identified NbALFAPE (for Peptide Elution) following a rational approach based on the NbALFA-ALFA peptide structure. In my opinion more information about the approach and the outcome is needed to justify publication in a scientific journal.

We agree with reviewer#2 that the approach to come to NbALFA^{PE} is a bit too short. However, as mentioned before, there is a patent application on this technology (which we indicate). Therefore, we won't be able to provide full information on the NbALFA^{PE}. The ALFA system here described and characterized in detail (more than almost any antibody regularly used in scientific publications) is already been used by dozens of beta testers around the scientific community. Therefore, we disagree that the sequence information of NbALFA^{PE} is vital to consider our data valid or of appropriate quality. To give the reader an idea about the approach followed in order to develop NbALFA^{PE}, we now expanded the corresponding paragraph (p.10, lines 22-29):

"To allow for an efficient competitive elution of ALFA-tagged target proteins, we intended to weaken NbALFA. For that, we followed a rational approach based on the NbALFA-ALFA peptide structure: NbALFA variants featuring single or combined amino acid exchanges in spatial proximity to the ALFA peptide were individually tested for their binding and elution properties. The successfully weakened binder NbALFA^{PE} (for Peptide Elution) carries several mutations with respect to NbALFA, thereby removing specific interactions of NbALFA with the ALFA peptide. As a consequence, the affinity of NbALFA^{PE} to a fusion protein harboring a C-terminal ALFA-tag was reduced to ~11 nM (Fig.S5). As intended, an NbALFA^{PE}-charged resin (ALFA Selector^{PE}) efficiently..."

4. DNA-PAINT experiments: The methods (p22, l30) suggest that conjugation of the docking strand to the nanobody was performed after fixation and permeabilization. Is this really how the experiment was done? If so, why is that? Is the product very unstable? In addition, details about the conjugation are missing. What chemistry was used for this?

We thank the reviewer for pointing out this misleading phrasing and clarified the method part accordingly. The nanobody was actually conjugated to the docking strand before and the whole protein-DNA construct was used for staining. Concerning the conjugation reaction, the azide-DNA docking strand was coupled via a maleimide-DBCO crosslinker to ectopic cysteine residues on the nanobody. We rewrote the corresponding methods section and now provide more details about the conjugation reaction. In order to improve clarity, we now collected all technical information regarding the DNA-PAINT in a combined paragraph "**DNA-labeling of NbALFA, DNA-PAINT imaging and data analysis**" (p.24). The former, combined paragraph describing all microscopical techniques has been adjusted accordingly and now focuses on epifluorescence and STED microscopy (p.23).

Other changes to the manuscript:

- We replaced the bar graph in Figure S2 by a scatter blot.
- We introduced a new source data file containing the raw data used for affinity determinations as well as raw data of graphs shown in figures 4b, S2, S5 and S6 according to Nature policies.
- In the section "Data availability", we now introduced a reference to the (new) source data file.
- We updated the author affiliations, author contributions and acknowledgements, citations and figure numberings.

Reviewers' comments:

Reviewer #1 (Remarks to the Author):

In this revised version, the authors replied satisfactorily to the requests of the referee's. However, I still have a minor favour to ask:

In the M&M section "Selection of specific sdAb clones": "The alpaca was immunised with the ALFA peptide.... ". We absolutely need slightly more explanations on how the animal was immunised: how meh peptide, which adjuvant, how many boosts. This is highly relevant as the immune system of camelids do normally NOT generate good immune response against short peptides.

Secondly, in the same paragraph: "From the eluted B cells an sdAb specific cDNA library was amplified". Here it would be nice to read how many B cells were retrieved using this method. Also, which PCR primers were used for the nested PCR? (Sequence). How large was the library? Which expression vector was taken ("96 clones were screened for expression of ALFA reactive sdAbs") How many of these were positive in ELISA ? (Some of these questions can be answered by citing the relevant articles). These questions are all relevant informations that should give the reader a sense of how this 'non-conform' library was generated and on the success of identifying target specific clones from such library.

Reviewer #2 (Remarks to the Author):

The authors address most of my concerns by explaining that knowing the full details of the new approach is not essential and that companies need to protect their IP in order to survive. I completely understand this last point, which is why most companies do not aim to publish papers in scientific journals.

Response to comment 1:

If the manuscript is just a "sound characterization of a novel epitope tag", I would never recommend it for Nature Communications. The design aspect is an important part of the story, as the authors also acknowledge themselves: "our paper has a meta-level that goes beyond this "simple" technical aspect: It describes a new approach allowing to develop affinity tags (and the respective binders) with predefined properties." As such, I completely disagree with their following statement: "We do not believe that it is essential for the described approach to fully understand the origin of the final ALFA tag sequence". The authors should just explain that they took an existing sequence and explicitly justify the two changes they made.

Response to comment 2:

In my opinion, you either make a product and sell it while protecting it, or you aim for a scientific publication, in which case you disclose your full approach. The Sigma-Aldrich #F-3156 example does not fly, since Sigma is not submitting papers characterizing their antibodies to Nature Communications.

In the end, I leave it to the editor to decide whether Nature Communications is a good platform for the characterization, but not full disclosure, of novel methods.

Point-by-point reply to the Reviewers' comments

Color code:

Black: Original comment by the Reviewer

Blue: Answer by authors

Green: Sections with changes implemented to the manuscript.

Reviewer #1 (Remarks to the Author):

In this revised version, the authors replied satisfactorily to the requests of the referee's. However, I still have a minor favour to ask:

In the M&M section "Selection of specific sdAb clones": "The alpaca was immunised with the ALFA peptide.... ". We absolutely need slightly more explanations on how the animal was immunised: how meh peptide, which adjuvant, how many boosts. This is highly relevant as the immune system of camelids do normally NOT generate good immune response against short peptides.

Secondly, in the same paragraph: "From the eluted B cells an sdAb specific cDNA library was amplified'. Here it would be nice to read how many B cells were retrieved using this method. Also, which PCR primers were used for the nested PCR? (Sequence). How large was the library? Which expression vector was taken ("96 clones were screened for expression of ALFA reactive sdAbs") How many of these were positive in ELISA ? (Some of these questions can be answered by citing the relevant articles). These questions are all relevant informations that should give the reader a sense of how this 'non-conform' library was generated and on the success of identifying target specific clones from such library.

Answer:

The number of retrieved B-cells has not been quantified. Judged from the amount of extracted total RNA (590ng) it is most likely in the range of 10^4 - 10^5 cells. Similarly, the complexity of the library has not been determined. As we intended to directly analyze a very low number of single clones, it was not required to aim at the highest possible library size or transformation efficiency. As mentioned in the manuscript, it is important to note that no further selection (like phage display etc.) was involved in the process, i.e., the cDNA obtained after reverse transcription and PCR amplification was directly cloned into a prokaryotic expression vector. From 96 clones analyzed by ELISA, 16 showed a signal >5-fold above background, which

were further sequenced. 13 of these clones could be grouped into two dominant sequence families. The other 3 clones were non-redundant.

Similar to phage-display, the outcome of the novel selection method (i.e. number of positive clones etc.) is strongly dependent on the specific antigen. We, e.g., have obtained between <10 and >90% (generally 30-70%) positive clones with the same technique using different target proteins. The number of positive clones obtained while screening for ALFA-reactive nanobodies is thus more at the lower end of the "normal" range – more or less as expected for a peptide antigen with rather "poor" antigenicity.

In order to give more background information on the immunization and screening protocols, we now introduced a new paragraph "Immunizations" in the methods section and modified the paragraph "Selection of specific sdAb clones by affinity enrichment of B lymphocytes".

They now read as follows:

Immunizations

Two alpacas were immunized 6 times with 0.5mg ALFA-peptide containing an additional N-terminal cysteine (CSRLEEELRRRLTE-Amide) conjugated to 1mg MBS-activated KLH (0.9mL + 0.9mL adjuvant). The first immunization (day 0) was done using complete Freund's adjuvant, for all following immunizations (days 14, 28, 42, 56 and 112) incomplete Freund's adjuvant was used. 100mL peripheral blood was taken from each animal 5 days after the last immunization and immediately incubated with 5000 IU/mL heparin (Sigma) to prevent clotting.

Selection of specific sdAb clones by affinity purification of B lymphocytes

1mL of T-Catch resin (IBA Lifesciences) was washed with B cell isolation buffer (PBS pH7.4, 1% BSA, 1mM EDTA) and incubated with saturating amounts of a TwinStrepTag-bdNEDDD8-ALFA fusion protein for 30min rolling at RT. The resin was cleared from excess bait protein by extensively washing with B cell isolation buffer. 100mL of blood sample was taken from alpaca immunized with ALFA peptide and immediately incubated with 5000 IU/mL heparin (Sigma) to prevent clotting. From the fresh blood (less than 4h past sampling) PBMCs were isolated using Ficoll-Paque PLUS (GE Healthcare). To remove residual serum, PBMCs were washed three times consecutively with B cell isolation buffer. PBMCs were passed three times over the antigen-loaded T-Catch resin before washing the resins with 10 CV B cell isolation buffer. Bound B cells were eluted from the resins by incubating with 2µM NEDP1^{40,41} for 30min at RT. Total RNA was extracted from the eluted B cells (590ng total)

and used for reverse transcription and PCR amplification as previously described²². The amplified sdAb library was directly cloned into a pQE80-derived prokaryotic expression vector in frame to an N-terminal His-bdSUMO-tag and a C-terminal FLAG-GFP-tag. 96 single clones were expressed in deep-well plates. After cell disruption, crude lysates were tested by ELISA for the presence of ALFA-reactive sdAbs. 16 clones showing signals >5-fold above background were sequenced and further characterized biochemically. 13 of these clones could be grouped into two independent sequence families with 10 and 3 clones, respectively. NbALFA belonged to the most abundant family of clones that biochemically behaved most favorable.

Reviewer #2 (Remarks to the Author):

The authors address most of my concerns by explaining that knowing the full details of the new approach is not essential and that companies need to protect their IP in order to survive. I completely understand this last point, which is why most companies do not aim to publish papers in scientific journals.

Response to comment 1:

If the manuscript is just a “sound characterization of a novel epitope tag”, I would never recommend it for Nature Communications. The design aspect is an important part of the story, as the authors also acknowledge themselves: “our paper has a meta-level that goes beyond this "simple" technical aspect: It describes a new approach allowing to develop affinity tags (and the respective binders) with predefined properties.” As such, I completely disagree with their following statement: “We do not believe that it is essential for the described approach to fully understand the origin of the final ALFA tag sequence”.

The general approach we were referring to does not include the selection of a *specific* peptide, but more generally, the *de-novo* design of an epitope tag based on our defined selection criteria (in contrast to most epitope tags being “side-products”). We believe that such approach (namely rationally equipping epitope tags with pre-defined characteristics) would work for a large number of completely different sequences. In fact, in our example, we "just" took a sequence from the literature that already nearly matched our selection criteria and only modified it slightly in order to perfectly match these criteria. This shows that there is no "magic knowledge" behind the peptide selection itself – besides fulfilling the described selection criteria, which we describe in the manuscript.

The authors should just explain that they took an existing sequence and explicitly justify the two changes they made.

We already mentioned in our manuscript that we indeed took an alpha-helical sequence described before and detail on the criteria for selection of this sequence. In our "Response to the Reviewers' comments", we in addition also explain the further changes introduced. In response to the new Reviewer's request, we now modified and expanded the section in question once again. It now reads:

The sequence of the minimal ALFA-tag (SRLEEEELRRRLTE; Fig.1a) is inspired by an artificial peptide (SRLEEEELRRRL) reported to form a stable α -helix in solution²³. It was selected based on the following criteria: i) It features a high alpha-helical content, ii) The sequence is absent in common eukaryotic model systems, iii) It is hydrophilic and neutral at physiological pH while retaining moderate hydrophobic surfaces and iv) It does not contain any primary amines that are modified by aldehyde-containing fixatives. The additional Thr-Glu (TE) dipeptide was added to the original peptide to neutralize its positive net charge and thus fully comply with the selection criteria defined above. To minimize any potential influence of neighboring secondary structures, the minimal ALFA-tag sequence was in addition framed by prolines (Fig.1a).

Response to comment 2:

In my opinion, you either make a product and sell it while protecting it, or you aim for a scientific publication, in which case you disclose your full approach. The Sigma-Aldrich #F-3156 example does not fly, since Sigma is not submitting papers characterizing their antibodies to Nature Communications.

In the end, I leave it to the editor to decide whether Nature Communications is a good platform for the characterization, but not full disclosure, of novel methods.

It might seem uncommon, but we actually do make nearly our complete technology freely available to the academic community AND at the same time try to launch a series of commercial products. This product line is intended for all scientists that may be unable to produce nanobody-derived products in their lab (or simply want to rely on validated products). The only element we indeed wish to protect is the sequence of NbALFA^{PE} (which

will be published in the course of our patent and thus also be made available to the academic community soon).

As it seems difficult to *convince* Reviewer2 regarding this point, we completely agree with the latter statement and leave it to the editor to judge on the significance of the ALFA system. In this context we would like to mention the large attention our manuscript has already gained on bioRxiv (<https://www.biorxiv.org/content/10.1101/640771v2>): Within only one week after publication our manuscript has reached the top 1% regarding its high attention score (Altmetric) when compared with all outputs of the same age published at bioRxiv, and even when compared to ALL research output evaluated by Altmetric. Already now (after ~3 weeks), our manuscript has been downloaded more than 2000 times (as full-text HTML or pdf). As a result, we were able to send out plasmids encoding NbALFA and transfection vectors encoding ALFA-tagged target proteins (for free) to dozens of interested users all over the world.

Changes implemented without direct connection to the Reviewers' comments:

- Figure 3 has been replaced by an updated version due to a mistake in the labeling of Fig. 3b ("E105" instead of the correct "D105")
- We corrected any inconsistency in naming of the TwinStrep-tag.